# Reconstruction and Secrecy under Approximate Distance Queries

**Shay Moran**
Department of Mathematics, Technion
Department of Computer Science, Technion
Department of Data and Decision Sciences, Technion
Google research

**Elizaveta Nesterova**
Department of Mathematics, Technion

## Abstract

Consider the task of locating an unknown target point using approximate distance queries: in each round, a reconstructor selects a reference point and receives a noisy version of its distance to the target. This problem arises naturally in various contexts—ranging from localization in GPS and sensor networks to privacy-aware data access—and spans a wide variety of metric spaces. It is relevant from the perspective of both the reconstructor (seeking accurate recovery) and the responder (aiming to limit information disclosure, e.g., for privacy or security reasons). We study this reconstruction game through a learning-theoretic lens, focusing on the rate and limits of the best possible reconstruction error. Our first result provides a tight geometric characterization of the optimal error in terms of the Chebyshev radius, a classical concept from geometry. This characterization applies to all compact metric spaces (in fact, even to all totally bounded spaces) and yields explicit formulas for natural metric spaces. Our second result addresses the asymptotic behavior of reconstruction, distinguishing between pseudo-finite spaces—where the optimal error is attained after finitely many queries—and spaces where the approximation curve exhibits a nontrivial decay. We characterize pseudo-finiteness for convex Euclidean spaces.

## 1 Introduction

In the *reconstruction game*, a reconstructor seeks to locate an unknown point $x^\star$ in a metric space $(X, \operatorname{dist}_X)$ using a sequence of approximate distance queries. In each round, the reconstructor selects a query point $q_t \in X$ and receives a response $\hat{d}_t$ that approximates the true distance $\operatorname{dist}_X(q_t, x^\star)$. The approximation is controlled by two error parameters: $\epsilon \geq 0$, which bounds the multiplicative error, and $\delta \geq 0$, which bounds the additive error. Specifically, the response satisfies $\hat{d}_t =_{\epsilon,\delta} \operatorname{dist}_X(q_t, x^\star)$, where

$$x =_{\epsilon,\delta} y \quad \text{means that} \quad x \leq (1+\epsilon)y + \delta \quad \text{and} \quad y \leq (1+\epsilon)x + \delta.$$

After a bounded number of such queries, the reconstructor outputs a guess $\hat{x} \in X$ and aims to minimize the reconstruction error $\operatorname{dist}_X(\hat{x}, x^\star)$.

This simple game arises in a wide range of natural scenarios. In *privacy-preserving data analysis*, it models the trade-off between utility and privacy: a responder must answer queries while protecting sensitive data, as in the foundational work of [Dinur and Nissim, 2003] that initiated the study of differential privacy[1]. In *computational geometry*, related questions arise when inferring geometric structures from noisy measurements [Disser and Skiena, 2017]. In *remote sensing*, satellites and sensors reconstruct physical information—such as terrain or atmospheric properties—from indirect and error-prone signals [Twomey, 1977]. Similar structures also appear in *learning theory*: for instance, hypothesis selection and distribution learning via statistical queries can be framed as reconstruction problems over suitable metric spaces.

---

[1]The reconstruction model studied by Dinur and Nissim uses counting (or linear) queries, but it is essentially equivalent to our model with distance queries. We elaborate on this connection at Example 1.

39th Conference on Neural Information Processing Systems (NeurIPS 2025).

The reconstruction game captures a natural tension between two objectives: recovering hidden information and limiting what can be revealed. From the reconstructor's perspective, the task is to approximate an unknown point from noisy distance measurements. This challenge arises in a variety of applications, including navigation, search-and-rescue, and remote sensing, where inference must be made under uncertainty. On the other side, the responder may wish to share useful information while restricting what can be inferred—whether for reasons of privacy, security, or resource constraints. This interplay between noisy access and limited disclosure makes the model relevant across several domains.

While this framework treats the reconstructor and the responder symmetrically, assigning equal roles to both players, our technical results are directed toward understanding the limits of what the reconstructor can achieve. We present two main contributions:

**Limit of optimal reconstruction (Theorem 2).** We characterize the optimal approximation error that the reconstructor can guarantee in the limit, as the number of queries tends to infinity. This error depends on the metric space and the approximation parameters $\epsilon$ and $\delta$, and our characterization applies to all totally bounded metric spaces. The result is expressed in terms of a classical geometric quantity: the Chebyshev radius. This limiting error plays a role analogous to the Bayes optimal error in statistical learning: it captures the best achievable performance in the presence of noise, regardless of the specific strategy or number of queries. Our result provides a geometric characterization and interpretation of this optimum in the context of the reconstruction game.

**Pseudo-finite Spaces (Theorem 6).** Beyond the limiting error, a central question concerns the rate at which this optimum is approached as a function of the number of queries. This question is inherently rich and depends delicately on the geometry of the space. It is analogous to the study of *learning curves* in statistical learning theory, which quantify how the performance of a learner improves with more data. We initiate the study of this question in our setting by identifying and analyzing a fundamental distinction between *pseudo-finite* spaces—where finitely many queries suffice to reach the optimum—and spaces where convergence is gradual. We show that this notion is already subtle and nontrivial, and we provide a characterization of pseudo-finiteness for convex Euclidean spaces.

## 1.1 Problem Setup and Main Results

We now define the *reconstruction game*, a formal interaction between two players: a *reconstructor* (RC) and a *responder* (RSP). The game takes place in a metric space $(X, \mathrm{dist}_X)$, where $X$ is the domain and $\mathrm{dist}_X : X \times X \to \mathbb{R}_{\geq 0}$ is a distance function. The interaction is governed by two error parameters: $\epsilon \geq 0$, controlling multiplicative distortion, and $\delta \geq 0$, controlling additive distortion.

---

**The Reconstruction Game**

The game is parameterized by $\epsilon, \delta \geq 0$, and is played over $T$ rounds in a metric space $(X, \mathrm{dist}_X)$. Each round $t = 1, \ldots, T$ proceeds as follows:

1. The reconstructor submits a query point $q_t \in X$.

2. The responder returns a value $\hat{d}_t$, which approximates the true distance to some secret point.

The responder must ensure that all answers given in the game remain jointly consistent with at least one point $x^\star$. That is: $(\exists x^\star \in X)(\forall t \leq T) : \hat{d}_t =_{\epsilon,\delta} \mathrm{dist}_X(q_t, x^\star)$, where $x =_{\epsilon,\delta} y$ means that $\hat{d}_t \leq (1+\epsilon)\mathrm{dist}_X(q_t, x^\star) + \delta$ and $\mathrm{dist}_X(q_t, x^\star) \leq (1+\epsilon)\hat{d}_t + \delta$.

At the end of the game, the reconstructor outputs a final guess $\hat{x}_T \in X$. The reconstruction error is defined as the worst-case distance to a consistent point: $\sup_{x^\star} \mathrm{dist}_X(\hat{x}_T, x^\star)$, where $x^\star$ ranges over all consistent points.

---

The reconstruction game studied in this work generalizes the task of determining the *sequential metric dimension* (SMD), which was originally introduced in the *noiseless* setting for graphs by Seager [2013]. The SMD captures the minimum number of exact distance queries required to identify an unknown point exactly, and has been studied in finite metric spaces induced by graphs [Bensmail

et al., 2020, Ódor and Thiran, 2021, Tillquist et al., 2023], with particular emphasis on the gap between sequential and static metric dimension.[2]

As another example, the counting-query model introduced by Dinur and Nissim [2003] in their foundational work can also be naturally viewed as a special case of our reconstruction game on the Boolean cube endowed with the Hamming metric:

**Example 1** (From counting queries to distance queries). *In the counting-query model [Dinur and Nissim, 2003], the dataset is a binary vector $D = (d_1, \ldots, d_n) \in \{0, 1\}^n$. At round $t = 1, \ldots, T$, the reconstructor chooses a subset $q_t \subseteq [n]$ and receives*

$$a_t = \sum_{i \in q_t} d_i + \eta_t, \qquad |\eta_t| \leq \delta.$$

*This game is not syntactically a metric-distance game, yet it is* equivalent *to our distance-query model on the Boolean cube with the Hamming metric, in the sense that counting queries and Hamming-distance queries simulate each other with at most a two-query overhead per round. Full details of the simulation appear in Appendix F.3, Example 30.*

**A Priori vs. A Posteriori Responder.** There are two natural variants of the reconstruction game, which differ in when the responder commits to the secret point.

In the *a priori* version, the responder selects a secret point $x^\star \in X$ at the beginning of the game and must answer all queries consistently with that fixed point. In contrast, the *a posteriori* version (which we adopt) allows the responder to wait until the end of the game when the reconstructor selects her guess $\hat{x}_T$, before selecting the secret point $x^\star$.

Note that for deterministic reconstructors, the a priori and a posteriori models are equivalent: any a posteriori responder can be simulated by an a priori one, simply by anticipating all queries of the reconstructor and precomputing a worst-case consistent point in advance. "Deterministic" here refers only to the absence of internal randomness and does not restrict adaptivity; a deterministic reconstructor may choose each query based on the entire interaction so far, whereas "non-adaptive" denotes the special case in which all queries are fixed in advance.

**Remark.** *We define the game using the a posteriori model because our results focus on the capabilities of the reconstructor in the* worst-case *setting. From this viewpoint, the most meaningful formulation is one where the responder is allowed maximal flexibility, making the task of reconstruction as difficult as possible.*

## 1.2 Optimal Reconstruction Distance

At each point in the game, the sequence of query–response pairs received so far determines a *feasible region*—the set of points in $X$ that are consistent with all previous answers under the error model. The size and geometry of this region reflect the remaining uncertainty about the secret point. From the reconstructor's perspective, the goal is to make this region as small as possible, ideally identifying a point that is close to every element in it. We measure the performance of a reconstruction strategy by the worst-case distance between the output $\hat{x}_T$ and any point in the feasible region. The key quantity we study is the optimal worst-case guarantee achievable by the reconstructor after $T$ queries, denoted by

$$\text{OPT}_X(T, \epsilon, \delta) := \inf_{\text{RC}} \sup_{\text{RSP}} \sup_{x \in \Phi_T} \text{dist}_X(\hat{x}_T, x). \tag{1}$$

Here, $\Phi_T \subseteq X$ is the *feasible region*—the set of points that remain consistent with the transcript (i.e., the sequence of queries and responses) $\{(q_t, \hat{d}_t)\}_{t=1}^T$

$$\Phi(\{q_i, r_i\}_{i=1}^T) := \left\{ x \in X \mid \text{for all } 1 \leq i \leq T: \ \text{dist}_X(x, q_i) =_{\epsilon, \delta} r_i \right\}. \tag{2}$$

The infimum ranges over all strategies employed by the reconstructor, and each such strategy is evaluated in the worst case: against the most adversarial responder strategy (subject to consistency), and with respect to the most distant feasible point. For randomized reconstructors, we interpret Equation (1) by replacing $\text{dist}_X(\hat{x}_T, x)$ with $\mathbb{E}[\text{dist}_X(\hat{x}_T, x)]$, where the expectation is over the

---

[2]The static metric dimension is the minimum number of reference points needed to uniquely determine any point in the space based on its distances to those references. It corresponds to the non-adaptive variant of our setting, where all queries are fixed in advance.

internal randomness of the reconstructor. For simplicity of presentation, we assume the reconstructor is deterministic; however, all of our results and proofs extend to the randomized setting.

Much of our focus will be on understanding how this function behaves as $T \to \infty$, and how it depends on the geometry of the underlying metric space $X$. Our first main result concerns the asymptotic quantity

$$\mathrm{OPT}_X(\epsilon, \delta) := \lim_{T \to \infty} \mathrm{OPT}_X(T, \epsilon, \delta),$$

which captures the best reconstruction error the reconstructor can guarantee in the limit, as the number of queries grows[3].

**Chebyshev Radius.** To characterize $\mathrm{OPT}_X(\epsilon, \delta)$, we rely on a classical geometric quantity called the *Chebyshev radius*, which captures how well a set can be enclosed by a ball. Let $(X, \mathrm{dist}_X)$ be a metric space, and let $\alpha > 0$ be a parameter. For any subset $S \subseteq X$, we denote its diameter by $\mathrm{diam}(S) := \sup_{x,y \in S} \mathrm{dist}_X(x,y)$. The *Chebyshev radius* of $S$, denoted $r(S)$, is defined as

$$r(S) := \inf_{x \in X} \sup_{y \in S} \left[ \mathrm{dist}_X(x,y) \right],$$

that is, the smallest radius for which some ball centered in $X$ contains all of $S$.

We will also rely on the following quantity that captures the worst-case relationship between sets of diameter at most $\alpha$ and the radius of their smallest enclosing ball:

$$\mathsf{e}_X(\alpha) := \sup_{S : \, \mathrm{diam}(S) \leq \alpha} r(S).$$

For example, in Euclidean space $(\mathbb{R}^n, \ell_2)$, it is known that for all $\alpha > 0$, $\mathsf{e}_X(\alpha) = \sqrt{\frac{n}{2(n+1)}} \cdot \alpha$, as shown, for instance, in Blumenthal [1970]'s monograph.

Before stating our first main result, we recall a standard notion from metric geometry. A metric space $(X, \mathrm{dist}_X)$ is said to be *totally bounded* if for every $r > 0$, there exists a finite cover of $X$ by balls of radius $r$. This is a common weakening of compactness that still ensures many desirable finiteness properties. As we will see in Section 2, this assumption is necessary for the theorem's conclusion; without it, the game can trivialize, allowing the responder to force an approximation error equal to the space's diameter.

---

**Tight Error via Chebyshev Radius**

**Theorem 2.** *Let $X$ be a totally bounded metric space. Then, for any $\epsilon, \delta \geq 0$,*

$$\mathrm{OPT}_X(\epsilon, \delta) = \mathsf{e}_X\big((2+\epsilon)\delta\big).$$

*Moreover, if the distance $(2+\epsilon)\delta$ is realized in $X$, i.e., there exist a pair of points at this distance, then*

$$\frac{1}{2}(2+\epsilon)\delta \leq \mathrm{OPT}_X(\epsilon, \delta) \leq (2+\epsilon)\delta.$$

---

This result expresses the limiting reconstruction error in terms of the function $\mathsf{e}_X(\cdot)$, which captures the worst-case Chebyshev radius over sets of bounded diameter. While the definition of $\mathsf{e}_X((2+\epsilon)\delta)$ may seem somewhat cryptic at first glance, it is often closely tied to the scale of noise introduced by the responder. Specifically, in many natural spaces, it holds that

$$\mathrm{OPT}_X(\epsilon, \delta) = \Theta\big((2+\epsilon)\delta\big).$$

This follows from the next general observation, which bounds the ratio between the Chebyshev radius and the diameter of a set in any metric space.

**Observation 3.** *In any metric space $(X, \mathrm{dist}_X)$ and every $\alpha > 0$ which is realized as a distance in the space, $\frac{1}{2}\alpha \leq \mathsf{e}_X(\alpha) \leq \alpha$. The upper bound follows because any set of diameter $\alpha$ can be trivially enclosed in a ball of radius $\alpha$. The lower bound holds because no ball of radius $r < \alpha/2$ can contain two points at distance $\alpha$.*

The bounds $\alpha/2$ and $\alpha$ are tight: they are attained by natural totally bounded metric spaces, as we will demonstrate through examples in Section 2.

---

[3]As shown in Appendix D, specifically in Claim 13, the function $\mathrm{OPT}_X(T, \epsilon, \delta)$ is monotonically non-increasing in $T$, so this limit always exists.

## 1.3 Excess Reconstruction Error

From a learning-theoretic perspective, the limiting error $\text{OPT}_X(\epsilon, \delta)$ plays a role analogous to the *Bayes optimal* error in statistical learning: it represents the best achievable performance under the constraints of the model. This motivates the study of the *excess reconstruction error*—the difference between the error achieved after $T$ queries and this asymptotic optimum: $\text{OPT}_X(T, \epsilon, \delta) - \text{OPT}_X(\epsilon, \delta)$. Understanding the rate at which this quantity decays as $T \to \infty$ is a natural next step. This question is generally quite challenging and depends intricately on the geometry of the underlying space. As a first step in this direction, our second main result focuses on a basic dichotomy: between spaces where convergence to the optimal error is trivial—i.e., achieved after finitely many queries—and all others. We formalize this notion through the following definition:

**Definition 4** (Pseudo-finite Spaces). *A metric space $(X, \text{dist}_X)$ is said to be $(\epsilon, \delta)$-pseudo-finite if there exists a finite constant $T_{X, \epsilon, \delta} < \infty$ such that*

$$\text{OPT}_X(T, \epsilon, \delta) = \text{OPT}_X(\epsilon, \delta) \quad \text{for all } T \geq T_{X, \epsilon, \delta}.$$

It is easy to see that any finite metric space is $(\epsilon, \delta)$-pseudo-finite for all values of $\epsilon, \delta \geq 0$: the reconstructor can simply query every point in the space; and no additional information can be obtained once all points have been queried. Another example of pseudo-finiteness is provided by finite-dimensional Euclidean spaces. The space $\mathbb{R}^n$ is $(0, 0)$-pseudo-finite[4] since the reconstructor can determine the exact location of the secret by querying $n + 1$ affinely independent points (see, e.g.[Tillquist et al., 2023]). In contrast, we will see in the next section an example of a totally bounded metric space that is not $(0, 0)$-pseudo-finite.

We now turn our attention to Euclidean spaces. Naturally, we begin with the simplest case: the real line. Despite its simplicity, the real line exhibits a nuanced pseudo-finiteness behavior that depends on the error parameters. In particular, pseudo-finiteness holds when there is no multiplicative noise, but breaks down as soon as any multiplicative distortion is allowed:

**Proposition 5** (Pseudo-finiteness of the real line). *Let $X = [0, 1] \subseteq \mathbb{R}$ equipped with the standard Euclidean metric. Then: (i) For every $\delta \geq 0$, the space $X$ is $(0, \delta)$-pseudo-finite. (ii) For every $\epsilon > 0$ and every $\delta \geq 0$, the space $X$ is not $(\epsilon, \delta)$-pseudo-finite.*

This proposition follows from our general result below (Theorem 6), but can also be derived more directly in this special case. When $\epsilon = 0$, the reconstructor can query one of the endpoints $q_1 \in \{0, 1\}$; the response confines the secret to an interval of length $2\delta$, and outputting its midpoint yields an error of at most $\delta$, which is optimal[5]. When $\epsilon > 0$, the responder can use a binary-search-like strategy to ensure that the feasible region always contains an interval of length strictly greater than $(2 + \epsilon)\delta$, thereby preventing the reconstructor from reaching the optimum in finitely many steps.

How about higher-dimensional Euclidean spaces—do they exhibit the same behavior as the real line with respect to pseudo-finiteness? Our second main result addresses this question for the class of convex subsets of Euclidean space. To state it, we recall that the *dimension* of a convex set $X \subseteq \mathbb{R}^n$ refers to the dimension of its affine span, i.e., the smallest affine subspace containing $X$. In higher dimensions, this nuanced behavior disappears: convex subsets of $\mathbb{R}^n$ with dimension at least two are never pseudo-finite, regardless of the values of $\epsilon$ and $\delta$, as long as they are sufficiently small compared to the diameter.

---

**Pseudo-Finiteness in Convex Euclidean Spaces**

**Theorem 6.** *Let $X \subset \mathbb{R}^n$ be a bounded convex set equipped with the Euclidean metric such that $\dim X > 0$ and let $\epsilon \geq 0$. Then, for all sufficiently small $\delta > 0$, the space $X$ is not $(\epsilon, \delta)$-pseudo-finite, except in the case where $\epsilon = 0$ and $\dim X = 1$.*

---

The proof of this result is surprisingly delicate. At a high level, one might expect that a responder could simply inject random noise into the true distances, thereby ensuring that the reconstructor improves only gradually over time. However, such a strategy does not suffice to rule out pseudo-finiteness: to do so, one must ensure that for every reconstructor strategy, the reconstruction error

---

[4]Note that noise plays an important role in this example: $\mathbb{R}^n$ is *not* pseudo-finite whenever the noise parameters are nonzero and $n \geq 2$.

[5]The optimality of the error $\delta$ on the interval, for sufficiently small $\delta$, follows from the fact that $\mathsf{e}_{[0,1]}(\delta) = \delta$ together with Theorem 2.

remains strictly larger than the optimal limit for any finite number of queries. This requires carefully calibrated noise that not only misleads the reconstructor but also guarantees that the resulting feasible region strictly contains a set of points forming an extremal body—one that achieves the maximal Chebyshev radius under a bounded diameter constraint.

In fact, the lower bound on $\mathrm{OPT}_X(\epsilon, \delta)$ established in Theorem 2 is implicitly used in proving Theorem 6, as it certifies the minimal size of the region that the responder must preserve.

**Remark.** *The proof of Theorem 6 provides two lower bounds on the convergence rate of $\mathrm{OPT}(T, \epsilon, \delta)$: exponential in $T$ for $\epsilon \neq 0$ and double-exponential for $\epsilon = 0$. On the upper-bound side, obtaining a matching rate for $\delta > 0$ appears nontrivial, and the optimality of the known lower bounds remains unclear.*

*In the purely multiplicative case $\delta = 0$, however, $\mathrm{OPT}_X(\epsilon, 0) = 0$, and a matching exponential upper bound follows from a standard grid-refinement argument: the reconstructor queries a uniform grid of fixed size (depending only on the dimension), selects the grid point with the smallest reported distance, then recenters a new fixed-size grid at that point and rescales to a smaller neighborhood. Iterating this geometrically shrinks the feasible region, yielding an exponential upper bound on $\mathrm{OPT}_X(T, \epsilon, 0)$.*

**Organization.** In the next section (Section 2), we analyze and discuss basic examples of the reconstruction game. In Section 3, we provide a high-level overview of the main technical ideas used in our proofs. For space reasons, the related work is deferred to Appendix A, which surveys relevant literature from learning theory, privacy, and geometry. The complete formal proofs are presented in Appendices B through E. Appendix F collects technical lemmas from geometry and topology and provides full proofs of the examples sketched in Section 2, along with additional examples that further clarify the game.

## 2 Examples

This section presents illustrative examples of the reconstruction game in a variety of metric spaces. These examples shed light on different aspects of the problem, including the necessity of the assumptions in our main theorems and the range of geometric behaviors that can arise. They also help clarify the role of total boundedness in Theorem 2, and lead naturally to an open question about pseudo-finite totally bounded spaces. In contrast to the following sections—which focus more heavily on Euclidean metric spaces in the context of Theorem 6—this section is technically lighter and features some more "exotic" spaces. Full proofs of the examples discussed here appear in Appendix F.3.

### 2.1 Total Boundedness in Theorem 2

The first main result (Theorem 2) characterizes the limiting reconstruction error in terms of the Chebyshev radius function $\mathsf{e}_X(\cdot)$, assuming that the metric space $X$ is totally bounded. The following examples illustrate that this assumption is essential: if total boundedness is lifted, even seemingly natural spaces allow the responder to prevent the reconstructor from obtaining any meaningful approximation—specifically, an error bounded away from zero, or even infinite.

**Example 7** (Unbounded Space: The Real Line). *We begin with a simple case: $\mathbb{R}$ with its standard Euclidean metric. This space is not bounded (and hence not totally bounded), and the responder can exploit its unboundedness to maintain extremely large feasible regions throughout the game. A formal proof is given in Appendix F.3.*

The previous example showed that in some unbounded metric spaces, such as $\mathbb{R}$, the responder can force the reconstruction error to be arbitrarily large. This naturally raises the question: could boundedness alone suffice for the conclusion of Theorem 2? That is, can we strengthen the theorem by replacing total boundedness with the weaker assumption of boundedness? The answer is negative:

**Example 8** (Bounded but Not Totally Bounded: Discrete Countable Space). *Consider the space $X = \mathbb{N}$, the set of natural numbers equipped with the discrete metric: $\mathrm{dist}_X(i, j) = 0$ if $i = j$, and $\mathrm{dist}_X(i, j) = 1$ otherwise. This space is bounded (diameter 1) but not totally bounded.*

*Now, note that even if the responder must be fully honest (i.e., $\epsilon = \delta = 0$), it can always answer $\hat{d}_t = 1$. This ensures that the feasible region after every round remains an infinite subset of $X$ in which all*

*points are pairwise at distance 1. Consequently, the responder can choose a consistent point of distance 1 from the point guessed by the reconstructor, yielding an approximation error equal to the diameter of the space.*

## 2.2 Pseudo-Finiteness

Although our main result about pseudo-finiteness focuses on convex Euclidean spaces, the phenomenon is more subtle in general metric spaces. In this section, we present three infinite metric spaces. Two of these spaces are $(\epsilon, \delta)$-pseudo-finite for all values of $\epsilon, \delta \geq 0$, while the third is not even $(0, 0)$-pseudo-finite. These examples highlight the diversity of possible behaviors in general metric spaces and motivate an open question concerning the structural nature of pseudo-finiteness in totally bounded spaces.

**Example 9** (Sparse Subsets of the Real Line). *Let $X = \{0\} \cup \{2^{2^n} : n \in \mathbb{N}\} \subset \mathbb{R}$ with the standard Euclidean metric. Then $X$ is $(\epsilon, \delta)$-pseudo-finite for every $\epsilon, \delta \geq 0$.*

*To see this, let the reconstructor begin by querying the point $q_1 = 0$. The response $\hat{d}_1$ yields a feasible region consisting of a finite subset of $X$, whose size is bounded by a constant $N(\epsilon, \delta)$ that depends only on the noise parameters (and not on the specific value of $\hat{d}_1$). This is because the set $X$, when viewed as a monotone sequence, grows asymptotically faster than any geometric progression. After this initial step, the reconstructor continues to query all points in the feasible region to identify an optimal approximation.*

The above example is unbounded. This raises the question of whether there exist bounded infinite spaces that are $(\epsilon, \delta)$-pseudo-finite for all $\epsilon, \delta$. The next example shows that the answer is yes.

**Example 10** (Countable Discrete Metric Space Revisited). *Recall the space $X = \mathbb{N}$ with the discrete metric: $\mathrm{dist}_X(x, y) = 0$ if $x = y$, and 1 otherwise. This space is bounded, with diameter 1. As previously discussed (see Example 8), we have $\mathrm{OPT}_X(\epsilon, \delta) = 1$ for all $\epsilon, \delta \geq 0$. Therefore, the reconstructor can achieve optimal performance without submitting any queries, simply by outputting any fixed point in the space. Thus, $X$ is $(\epsilon, \delta)$-pseudo-finite for all $\epsilon, \delta \geq 0$.*

These two examples motivate the following open question: can similar behavior occur in totally bounded spaces?:

**Open Question 11.** *Let $X$ be a totally bounded metric space. Are the following two statements equivalent? (i) $X$ is finite. (ii) $X$ is $(\epsilon, \delta)$-pseudo-finite for all $\epsilon, \delta \geq 0$.*

We conclude this section by presenting a totally bounded metric space that is not $(0, 0)$-pseudo-finite:

**Example 12** (Infinite binary strings). *Let $X = \{0, 1\}^{\mathbb{N}}$ be the space of infinite binary sequences, equipped with the standard ultrametric,[6] defined by $d(\alpha, \beta) = 2^{-j}$, where $j$ is the first index at which $\alpha_j \neq \beta_j$. Then $X$ is a compact metric space that is **not** $(0, 0)$-pseudo-finite. The proof appears in Example 28.*

# 3 Technical Overview

In this section, we outline the key ideas behind the proofs of Theorem 2 and Theorem 6; complete proofs are deferred to Appendices C and E. To keep the exposition focused on the central arguments, we omit technical complications arising from cases where suprema or infima are not attained. These can be handled with standard limiting arguments but would introduce additional notation and obscure the main ideas.

## 3.1 Proof of Theorem 2

We begin by recalling the core assertion of Theorem 2. It characterizes the optimal reconstruction error $\mathrm{OPT}_X(\epsilon, \delta)$ in terms of the geometry of the metric space and the noise parameters $\epsilon$ and $\delta$. Specifically, it asserts that $\mathrm{OPT}_X(\epsilon, \delta)$ equals the maximum Chebyshev radius among all subsets of $X$ with diameter at most $(2 + \epsilon)\delta$:

$$\mathrm{OPT}_X(\epsilon, \delta) = \mathrm{e}_X\big((2 + \epsilon)\delta\big).$$

---

[6]This metric satisfies the *ultrametric inequality*: $d(x, z) \leq \max\{d(x, y), d(y, z)\}$, which is stronger than the standard triangle inequality. It implies, for instance, that all triangles are isosceles with the two longer sides equal.

To prove this, we begin by analyzing an idealized setting in which the reconstructor is allowed to query all points in the space. Of course, this is unrealistic in infinite spaces—but it serves as a useful thought experiment for understanding the limits of reconstruction.

Each query-answer pair $(q, r)$ determines a *feasible region* $\Phi(\{q, r\})$, which consists of all points whose noisy distances to $q$ are $(\epsilon, \delta)$-indistinguishable from $r$. The intersection of all these regions gives the overall feasible region of the interaction, denoted by $\Phi := \Phi(\{q, r_q\}_{q \in X})$.

**Upper Bound.** In the idealized case where *all* points in the space are queried, a simple yet insightful argument shows that the diameter of the feasible region $\Phi$ is at most $(2 + \epsilon)\delta$. Indeed, for any two points $A, B \in \Phi$, since $B$ was queried and $A$ remained feasible, the reported noisy distance must not exceed $\delta$, and therefore the true distance $\mathrm{dist}_X(A, B)$ cannot exceed $(2 + \epsilon)\delta$. By letting the reconstructor output the Chebyshev center of $\Phi$, the reconstruction error is at most $\mathsf{e}_X\left((2 + \epsilon)\delta\right)$.

When only finitely many queries are allowed, however, the reconstruction error can be significantly larger than in the idealized case; as shown in Section 2.1, there exist spaces in which this discrepancy is arbitrarily large.

Nevertheless, if the metric space $X$ is *totally bounded*, the reconstructor can approximate the idealized strategy arbitrarily well: by querying all points in a sufficiently dense finite cover, one ensures that the feasible region has diameter arbitrarily close to $(2 + \epsilon)\delta$. Such a finite cover exists by definition: a metric space is totally bounded if, for every $\alpha > 0$, it admits a finite $\alpha$-cover—that is, a finite subset such that every point in the space lies within distance $\alpha$ of some point in the cover. Denote by $N_\alpha$ the number of points in an $\alpha$-cover of the metric space $X$. As illustrated in Figure 2, after $N_\alpha$ queries the reconstructor can guarantee that the diameter of the feasible region is less than $(2 + \epsilon)\delta + \alpha'$, where $\alpha' = ((1 + \epsilon)^2 + 1)\alpha$. Hence, by outputting the Chebyshev center of the feasible region, the reconstructor ensures that the worst-case error after $N_\alpha$ queries is at most $\mathsf{e}_X((2 + \epsilon)\delta + \alpha')$, by the definition of $\mathsf{e}_X(\beta)$ as the maximum Chebyshev radius over all subsets of $X$ with diameter at most $\beta$.

It might be tempting to conclude that we are done, since the function $\mathsf{e}_X$ appears to be continuous. However, this inference is, in general, false: for arbitrary metric spaces, $\mathsf{e}_X$ need not be continuous. For instance, in finite metric spaces the function $\mathsf{e}_X$ is *not* continuous.

On the other hand, it can be shown that for totally bounded metric spaces the function $\mathsf{e}_X$ is right-continuous, which is sufficient for establishing the desired upper bounds. Nevertheless, proving right-continuity remains nontrivial in general: there exists a non–totally bounded metric space for which the corresponding function $\mathsf{e}_X$ fails to be right-continuous (see Example 29 in Appendix F.3).

We prove that the function $\mathsf{e}_X$ is right-continuous for totally bounded metric spaces using the theory of *hyperspaces*. Namely, given a metric space $X$, one considers the space of (nonempty) *compact* subsets of $X$, denoted $\mathcal{K}(X)$, equipped with metrics induced by the metric on $X$. The most standard choice is the *Hausdorff metric*: for subsets $S_1, S_2 \subseteq X$,

$$d_H(S_1, S_2) = \max\left\{ \sup_{x \in S_1} \inf_{y \in S_2} \mathrm{dist}_X(x, y), \ \sup_{y \in S_2} \inf_{x \in S_1} \mathrm{dist}_X(x, y) \right\}.$$

A variety of classical results are known for $(\mathcal{K}(X), d_H)$; for instance, when $X$ is compact, the hyperspace $\mathcal{K}(X)$ is compact as well. This is a classical fact in metric topology; see, e.g., Illanes and Jr. [1999][Theorem 3.5].

Both the diameter and the Chebyshev radius of a set are continuous functions on $(\mathcal{K}(X), d_H)$; this follows by bounding their variation in terms of the Hausdorff distance (see the detailed argument in Appendix F.1). Together with the compactness of $(\mathcal{K}(X), d_H)$, this yields, via a compactness argument, that $\mathsf{e}_X$ is right-continuous for compact metric spaces. For a totally bounded metric space $X$, in turn, one can show that $\mathsf{e}_X = \mathsf{e}_{\hat{X}}$, where $\hat{X}$ denotes the completion of $X$. Since the completion of a totally bounded metric space is compact by the classical Heine–Borel characterization for metric spaces (compact $\Leftrightarrow$ complete and totally bounded), it follows that $\mathsf{e}_X$ is right-continuous for totally bounded metric spaces as well. All techniques and formal proofs for the right-continuity of $\mathsf{e}_X$ are presented in Appendix F.1. The full proof of the upper bound appears in Appendix C.2.

**Lower Bound.** The crucial observation is that at the beginning of the game, the responder may select *any* subset $S \subset X$ of diameter at most $(2 + \epsilon)\delta$, and maintain the invariant $S \subseteq \Phi$ throughout

the interaction: In response to each query $q$, the responder identifies a point $S_{\min} \in S$ that minimizes the distance to $q$, and returns the perturbed value

$$r := (1 + \epsilon) \cdot \operatorname{dist}_X(q, S_{\min}) + \delta.$$

A simple calculation, which relies only on the triangle inequality, shows that every $s \in S$ satisfies $\operatorname{dist}_X(q, s) =_{\epsilon, \delta} r$, and hence $S$ remains feasible.

After the interaction concludes, given the reconstructor's final guess, the responder can choose a secret point at distance no less than $r(S)$ inside $S \subseteq \Phi$. This ensures that no reconstructor can guarantee an error smaller than $e_X((2 + \epsilon)\delta)$. A precise description of the responder strategy that preserves an extremal set is presented in Appendix C.1.

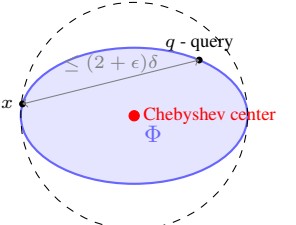

Figure 1: Feasible region $\Phi$ (blue) of the idealized case

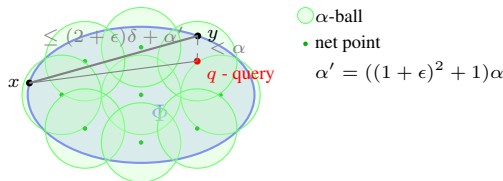

Figure 2: Feasible region $\Phi$ (blue) of the finite interaction

## 3.2 Proof of Theorem 6

We now turn to the proof of Theorem 6, which establishes a dichotomy for pseudo-finiteness in bounded convex subsets of Euclidean space. Specifically, the theorem states that a bounded convex set $X \subset \mathbb{R}^n$ is $(\epsilon, \delta)$-pseudo-finite if and only if $\dim(X) = 1$ and $\epsilon = 0$. In all other cases—namely, when $\dim(X) > 1$ or $\epsilon > 0$—the reconstruction error cannot reach its optimal value in finitely many steps.

One might hope to prove non-pseudo-finiteness by designing a responder strategy that gradually shrinks the feasible region—e.g., by adding uniform random noise to the true distance in each response. However, this naive approach fails to guarantee the desired behavior: in particular, it does not ensure that the reconstruction error remains strictly greater than the optimum $\operatorname{OPT}(\epsilon, \delta)$ at all finite $T$. In fact, such strategies may lead to convergence toward a strictly smaller value, and are therefore not optimal for the responder.

To overcome this, our proof explicitly constructs a responder strategy that, at every round, ensures the feasible region contains a subset guaranteeing that the reconstruction error remains strictly larger than the optimum $\operatorname{OPT}(\epsilon, \delta) + \alpha_T$, where $\alpha_T > 0$ depends only on the number of rounds.

This mirrors the lower-bound strategy used in Theorem 2, where the responder preserved an extremal set to ensure the Chebyshev radius never fell below $\operatorname{OPT}(\epsilon, \delta)$. However, to prove non-pseudo-finiteness, it is not sufficient to preserve a region whose radius merely equals the optimum. Instead, we must ensure that the feasible region's Chebyshev radius remains *strictly greater* than the limiting value for all finite $T$.

To accomplish this, our strategy preserves an $\alpha$-neighborhood of the vertices $\{x_i\}_{i=0}^n$ of some regular simplex $\Delta$[7], denoted $\Delta_\alpha$, where $\alpha > 0$ depends only on the number of rounds $T$. Formally, the $\alpha$-neighborhood of the simplex $\Delta$ with vertices $\{x_i\}_{i=0}^n$ is defined as

$$\Delta_\alpha := \cup_{i=0}^n B(x_i, \alpha), \qquad B(x_i, \alpha) \text{ denotes the Euclidean ball of radius } \alpha \text{ centered at } x_i. \quad (3)$$

This ensures that the feasible region contains a regular simplex of diameter $(2 + \epsilon)\delta + \sqrt{\frac{2(n+1)}{n}}\alpha$, which in turn implies that its Chebyshev radius is at least $\operatorname{OPT}(\epsilon, \delta) + \alpha$.

Our strategy proceeds as follows. Assume that at round $t$ the feasible region already contains an $\alpha_t$-neighbourhood of a regular simplex $\Delta$. Upon receiving the next query $q_t$, we pick a radius $\alpha_{t+1}$

---

[7]Throughout this work, by a "simplex" we usually mean the set of its vertices—that is, $n + 1$ affinely independent points in the Euclidean space $X$ of dimension $n$.

determined solely by $t+1, \epsilon, \delta$; then we reply with an appropriate noisy distance $r_t$ and, if necessary, replace $\Delta$ by a new extremal simplex $\Delta'$ so that

$$(\Delta')_{\alpha_{t+1}} \subset (\Delta)_{\alpha_t} \quad \text{and} \quad (\Delta')_{\alpha_{t+1}} \subset \text{(updated feasible region)}.$$

This step is then repeated indefinitely, keeping the Chebyshev radius strictly above OPT. Consequently, $X$ is not pseudo-finite.

The main challenge is to provide a *uniform* lower bound on $\alpha_t$ that depends only on the round $t$, and not on the specific query $q_t$. We note in passing that it is relatively easy to give a bound on $\alpha_{t+1}$ that depends on both $t$ and the query $q_t$; however, such a bound is insufficient for our purposes, as it does not yield a general lower bound on $\text{OPT}_T$ valid for all reconstructor strategies, which is essential for ruling out pseudo-finiteness.

On the other hand, finding the uniform bound requires handling each query type with care, since for some queries it is easy to obtain a sufficiently large neighborhood of some simplex $\Delta'$ contained within the neighborhood of the previous one, while for others it requires a more delicate geometric argument.

**Determining the Maximal Surviving Neighborhood.** To address the challenges above, we ask: under what conditions does there exist an answer that the responder can give to the query $q$ such that the $\alpha$-neighborhood (see Eq. (3)) of $\Delta$ remains entirely within the feasible region?

The answer is as follows: there exists such a response if and only if $r_q^{\min}(\Delta_\alpha) \le r_q^{\max}(\Delta_\alpha)$, where $r_q^{\min}$ places the farthest point of $\Delta_\alpha$ on the outer boundary of the feasible region, and $r_q^{\max}$ places the nearest point on the inner boundary (see Fig. 4, Fig. 3).

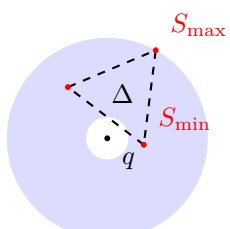

Figure 3: $\Phi(q, r^{\min})$ (blue)

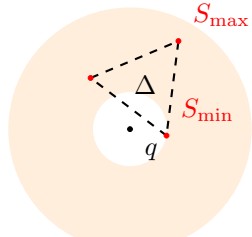

Figure 4: $\Phi(q, r^{\max})$ (orange)

The larger the radius $\alpha$ of the neighborhood $\Delta_\alpha$, the smaller the gap $r^{\max} - r^{\min}$ becomes. Solving the equation $r^{\max}(\Delta_\alpha) - r^{\min}(\Delta_\alpha) = 0$ for $\alpha$ yields the exact value $\alpha^\star(\Delta, q)$ of the largest surviving neighborhood upon querying $q$. It is useful to view the quantity $\alpha^\star(*, q)$ as a function on the space of regular simplexes. The derivation of the exact formula for $\alpha^\star$ is presented in Appendix D.

**Additive–Only vs. Multiplicative Noise.** Both responder strategies—additive-only and mixed-noise—rely on the same principle: for a fixed simplex $\Delta$ and a target neighborhood radius $\alpha_{t+1}$, together with the neighborhood radius $\alpha_t > \alpha_{t+1}$ from the previous round, we partition the space into $(\Delta, \alpha_{t+1})$-*good* and *bad* regions. A query point $q$ is called *good* if there exists a response that preserves a neighborhood of radius at least $\alpha_{t+1}$ of $\Delta$ within the feasible region; equivalently, if the maximal surviving neighborhood satisfies $\alpha^\star(\Delta, q) \ge \alpha_{t+1}$. Otherwise, $q$ is *bad*.

When $q$ is good, the responder can maintain the $\alpha_{t+1}$-neighborhood of the current simplex $\Delta$ inside the feasible region. The critical difference between regimes arises when $q$ is bad. In this case we should find another regular simplex $\Delta'$ in the $\alpha_t$-neighborhood of $\Delta$ such that the point $q$ is now $(\Delta', \alpha_{t+1})$-good. In the multiplicative case ($\epsilon > 0$), the responder can *translate* the simplex $\Delta$ slightly away from the query point $q$ to ensure that $\alpha^\star(\Delta', q) \ge \alpha_{t+1}$ and that $\Delta'_{\alpha_{t+1}} \subset \Delta_{\alpha_t}$.

In contrast, when $\epsilon = 0$, translations of $\Delta$ within its $\alpha_t$-neighborhood do not substantially change $\alpha^\star(*, q)$. To achieve a significant increase in $\alpha^\star(*, q)$ in this case, we *rotate* the simplex $\Delta$. To do this successfully, the rotated simplex must preserve the identity of the closest and farthest points from the query, which requires a careful geometric analysis. In Appendix F.2, we develop the tools necessary to carry out this strategy. In dimensions $n \ge 2$, rotations allow us to maintain a surviving neighborhood indefinitely. In one dimension, however—where nontrivial rotations are not possible—this strategy fails for a good reason: one-dimensional intervals are pseudo-finite.

## Acknowledgments and Disclosure of Funding

We thank Nikita Gladkov for insightful discussions related to the problems studied in this work.

Shay Moran is a Robert J. Shillman Fellow; he acknowledges support by ISF grant 1225/20, by BSF grant 2018385, by Israel PBC-VATAT, by the Technion Center for Machine Learning and Intelligent Systems (MLIS), and by the the European Union (ERC, GENERALIZATION, 101039692). Views and opinions expressed are however those of the author(s) only and do not necessarily reflect those of the European Union or the European Research Council Executive Agency. Neither the European Union nor the granting authority can be held responsible for them.

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

## A  Related Work

We organize the discussion into two parts: research that focuses on the *responder's perspective*, and research that centers on the *reconstructor's perspective*. In both cases, the relevant literature is vast, so we focus on works most closely related to the questions studied in this paper.

**The Responder's Perspective.**  The reconstruction game is closely related to problems studied in *privacy-preserving data analysis*, where the goal is to answer queries on a sensitive dataset while limiting what an adversary can infer [Dwork et al., 2006]. The foundational work of Dinur and Nissim [2003] initiated this line of research by showing that approximate answers to too many counting queries enable the reconstruction of a large fraction of the database. Their model uses counting queries on binary datasets, which are essentially equivalent to Hamming distance queries on the Boolean cube $\{\pm 1\}^n$. This connection is illustrated in Example 1.

Subsequent works have sharpened and generalized this reconstruction viewpoint. Notably, Dwork et al. [2007], Dwork and Yekhanin [2008], and Haitner et al. [2022] provided refined attacks and bounds under weaker assumptions. More recently, Balle et al. [2022] and Cummings et al. [2024] proposed formal definitions of *reconstruction robustness* that relate privacy guarantees to the attacker's ability to reconstruct sensitive data. Recent work by Cohen et al. [2025] further explores the foundations of reconstruction attacks, proposing a new definitional framework—Narcissus Resiliency—and uncovering connections to Kolmogorov complexity and classical notions such as differential privacy.

Surveys such as Dwork et al. [2017] provide a comprehensive overview of privacy attacks and defenses, including reconstruction. We also note the classical work of Erdos and Renyi [1963], which (in disguise) studies a version of the reconstruction problem on the Hamming cube in the noiseless setting.

**The Reconstructor's Perspective.**  Our work primarily studies the problem from the perspective of the *reconstructor*, who seeks to locate a hidden point using approximate distance queries. Related problems have been studied under several guises. A classic formulation is the *metric dimension* of a graph [Harary and Melter, 1976, Slater, 1975, Tillquist et al., 2023], which asks for the smallest set of vertices such that all other vertices are uniquely identified by their distances to this set. This corresponds to an *oblivious* version of the reconstruction game, where the reconstructor must submit all queries in advance.

A more sequential variant, closer to our setting, is the *sequential metric dimension* [Seager, 2013, Bensmail et al., 2020, Ódor and Thiran, 2021], which measures the number of adaptive queries needed to identify an unknown point. These works mostly consider noiseless settings on finite graphs. In contrast, our work allows noisy responses, considers general metric spaces, and studies the rate of convergence as a function of the number of queries.

The general formulation of locating a hidden point via distance queries has also appeared in applied contexts. For instance, the problem of reconstructing a physical quantity from noisy measurements arises in *remote sensing*, including terrain mapping and atmospheric profiling. Classic references include Twomey [1977] and Rodgers [2000], which formulate and analyze such problems as inverse problems under uncertainty. While much of this literature is algorithmic or statistical, our work provides a geometric and learning-theoretic view that complements these perspectives.

## B  General notation and basic facts

Let us remind the setup of the game and important concepts used throughout the proofs.

We work in a metric space $(X, \mathrm{dist}_X)$. The interaction lasts for a fixed number of rounds $T$, labeled $t = 1, 2, \ldots, T$. In round $t$, the reconstructor selects a query point $q_t \in X$. The responder then returns a real number $r_t$ that represents a noisy distance from $q_t$ to an as-yet-unspecified target, with multiplicative parameter $\epsilon \geq 0$ and additive parameter $\delta \geq 0$.

Formally, the reply must satisfy

$$\mathrm{dist}_X(x, q_i) \leq (1 + \epsilon)r_i + \delta,$$
$$r_i \leq (1 + \epsilon)\mathrm{dist}_X(x, q_i) + \delta$$

to at least one point $x \in X$. In other words, after each answer the set

$$\Phi(\{q_i, r_i\}_{i=1}^t) := \left\{ x \in X \mid \text{ for all } 1 \le i \le t: \begin{array}{l} \text{dist}_X(x, q_i) \le (1+\epsilon)r_i + \delta, \\ r_i \le (1+\epsilon)\text{dist}_X(x, q_i) + \delta \end{array} \right\}$$

is guaranteed to be non-empty. This set for the round $T$ is called the *feasible region*. In the end of the game the reconstructor outputs a guess point $\hat{x}_T$, and then the responder commits to a target point, choosing any $x^\star \in \Phi_T$.

The aim of the reconstructor is to minimize the distance $\text{dist}_X(x^\star, \hat{x}_T)$, and of the responder is to maximize it.

Let us denote the set of all reconstructors which play game for $T$ rounds by $\text{RC}_T$, and the set of responders by $\text{RSP}_T$. The final guess of the reconstructor $\mathcal{R} \in \text{RC}_T$ we will denote by $\hat{x}^{\mathcal{R}}$ and the output secret point of the responder $\mathcal{A} \in \text{RSP}_T$ by $x_{\mathcal{A}}^\star$. As recalled in Equation 1 the optimal error must be

$$\text{OPT}_X(T, \epsilon, \delta) := \inf_{\text{RC}} \sup_{\text{RSP}} \sup_{x \in \Phi_T} \text{dist}_X(\hat{x}_T, x).$$

**Claim 13** (Monotone error in $T$). *The function $T \mapsto \text{OPT}_X(T, \epsilon, \delta)$ is non-increasing.*

*Proof of Claim 13.* We show that for every $T \ge 0$, the function $T \mapsto \text{OPT}_X(T, \epsilon, \delta)$ is non-increasing; that is,

$$\text{OPT}_X(T+1, \epsilon, \delta) \le \text{OPT}_X(T, \epsilon, \delta).$$

Fix any $\alpha > 0$. By the definition of $\text{OPT}_X(T, \epsilon, \delta)$, there exists a reconstructor $\mathcal{R}$ that, after $T$ queries, guarantees an error at most $\text{OPT}_X(T, \epsilon, \delta) + \alpha$ against any responder.

Now consider a new reconstructor $\mathcal{R}'$ for $T+1$ rounds, which simulates $\mathcal{R}$ for the first $T$ queries, and then issues an arbitrary "dummy" query at round $T+1$, ignores the response, and simply outputs the same guess $\hat{x}_{T+1} := \hat{x}_T$ that $\mathcal{R}$ would have produced after $T$ rounds.

Since the feasible region after $T+1$ queries is always contained in the feasible region after $T$ queries, and since the final guess remains the same, the reconstruction error of $\mathcal{R}'$ is at most $\text{OPT}_X(T, \epsilon, \delta) + \alpha$ for any responder.

As this holds for every $\alpha > 0$, it follows that

$$\text{OPT}_X(T+1, \epsilon, \delta) \le \text{OPT}_X(T, \epsilon, \delta),$$

as required. $\qquad\square$

## C  Proof of Theorem 2

In this section, we present the full proof of Theorem 2. The proof is based on geometric notions such as the Chebyshev radius and the diameter of a set, together with a fundamental invariant of a metric space: the maximal radius of an enclosing ball over all subsets of bounded diameter.

We begin by recalling the relevant definitions. For a subset $S \subseteq X$, the *Chebyshev radius* of $S$, denoted $r(S)$, and the *diameter* of $S$, denoted $\text{diam } S$, are defined by

$$r(S) = \inf_{q \in X} \sup_{x \in S} \text{dist}_X(x, q), \qquad \text{diam } S = \sup_{x,y \in S} \text{dist}_X(x, y).$$

The supremum and infimum of a set of real numbers $\mathcal{A} \subset \mathbb{R}$ serve the same purpose as the maximum and minimum. The key difference is that the supremum or infimum may not be attained by any element of $\mathcal{A}$. In such cases, one can approximate it by a sequence $\{a_i\}_{i \in \mathbb{N}} \subseteq \mathcal{A}$ satisfying

$$\lim_{i \to \infty} a_i = \sup \mathcal{A} \quad \text{or} \quad \lim_{i \to \infty} a_i = \inf \mathcal{A}.$$

An important invariant used in the proof—intuitively, the maximal radius of an enclosing ball among all subsets of $X$ with diameter at most $\alpha$—is formally defined by

$$\mathbf{e}_X(\alpha) := \sup_{\substack{S \subseteq X \\ \text{diam}(S) \le \alpha}} r(S).$$

In some cases, this supremum is attained; when that happens, we refer to the corresponding subset $S \subset X$ as *extremal*. In general, even when the supremum is not attained, we may consider a sequence of subsets $\{S_m\}_{m \in \mathbb{N}}$ of bounded diameter, $\operatorname{diam} S_m \leq \alpha$, whose Chebyshev radii converge to the supremum:

$$\lim_{m \to \infty} r(S_m) = \mathsf{e}_X(\alpha).$$

The statement of Theorem 2 consists of two parts: an exact (tight) expression for OPT in terms of the function $\mathsf{e}_X$, and upper and lower bounds on $\mathsf{e}_X$.

We begin with the proof of the first part.

**Theorem 2** (First part). *Let $X$ be a totally bounded metric space. Then, for any $\epsilon, \delta \geq 0$,*

$$\mathrm{OPT}_X(\epsilon, \delta) = \mathsf{e}_X\big((2+\epsilon)\delta\big).$$

*Proof.* To prove the equality, we need to establish both directions:

$$\mathrm{OPT}_X(\epsilon, \delta) \geq \mathsf{e}_X\big((2+\epsilon)\delta\big) \quad \text{and} \quad \mathrm{OPT}_X(\epsilon, \delta) \leq \mathsf{e}_X\big((2+\epsilon)\delta\big).$$

*Lower bound*. To show that the optimal error is *at least* $\mathsf{e}_X((2+\epsilon)\delta)$, it suffices to construct responder strategies that guarantee a reconstruction error arbitrarily close to this value.

Although the supremum in the definition of $\mathsf{e}_X$ may not be attained by any single set, we can approximate it by a sequence of sets $\{S_m\}$ with $\operatorname{diam} S_m \leq (2+\epsilon)\delta$ and $r(S_m) \longrightarrow \mathsf{e}_X((2+\epsilon)\delta)$. For each such set, we define a responder strategy that preserves $S_m$ inside the feasible region, thereby ensuring that the reconstructor cannot achieve error smaller than $r(S_m)$. Taking the limit yields the desired lower bound.

*Upper bound*. To establish that the optimal error does not exceed $\mathsf{e}_X((2+\epsilon)\delta)$, we construct a sequence of reconstruction strategies, each using a query set of size $T_n$, such that the corresponding error remains within $\mathsf{e}_X((2+\epsilon)\delta) + \alpha_n$, where $\alpha_n \to 0$.

Since $X$ is totally bounded, for any precision level $\alpha > 0$, there exists a finite set $T_\alpha \subset X$ that forms an $\alpha$-cover of the space. After querying every point in such a cover, a feasible region has diameter smaller than $(2+\epsilon)\delta + \big((1+\epsilon)^2 + 1\big)\alpha$, and hence

$$\mathrm{OPT}(T_\alpha, \epsilon, \delta) \leq \mathsf{e}_X\big((2+\epsilon)\delta + \big((1+\epsilon)^2 + 1\big)\alpha\big).$$

The remaining step is to show that the function $\mathsf{e}_X$ is right-continuous, i.e.,

$$\mathsf{e}_X\big((2+\epsilon)\delta + \alpha'\big) \xrightarrow[\alpha' \to 0]{} \mathsf{e}_X\big((2+\epsilon)\delta\big),$$

which requires general machinery from topology—specifically, endowing the collection of compact subsets of $X$ with a natural metric that measures how far these subsets are from each other within $X$. This part of the proof is deferred to Appendix F.1.

### C.1  Lower bound via extremal sets

As mentioned earlier, the supremum

$$\mathsf{e}_X\big((2+\epsilon)\delta\big) := \sup_{\substack{S \subseteq X \\ \operatorname{diam}(S) \leq (2+\epsilon)\delta}} r(S)$$

plays the role of a maximum, although it may not actually be attained. In such cases, we simulate extremal sets—that is, sets that would attain this maximum—by considering approximately extremal sets: a sequence $\{S_m\}_{m \in \mathbb{N}}$ satisfying

$$r(S_m) \xrightarrow[m \to \infty]{} \mathsf{e}_X(\alpha), \qquad \operatorname{diam} S_m \leq (2+\epsilon)\delta.$$

For any $m \in \mathbb{N}$, define a responder strategy that, given any query $q \in X$, replies with

$$r_q := (1+\epsilon) \inf_{s \in S_m} \operatorname{dist}_X(q, s) + \delta.$$

Here, the infimum plays the role of a minimum; so if the minimum is attained at some point $B \in S_m$, this strategy effectively places $B$ on the boundary of the feasible region (see Figure 4).

Let us elaborate. We will show that for any point $s \in S_m$, the response satisfies

$$r_q \leq (1 + \epsilon)\text{dist}_X(q, s) + \delta,$$

and then, using the triangle inequality, we will obtain the reverse bound,

$$r_q \geq (1 + \epsilon)\text{dist}_X(q, s) - \delta.$$

The inequality $r_q \leq (1+\epsilon)\text{dist}_X(q, s) + \delta$ follows directly from the definition of the infimum, which represents the minimal possible distance:

$$(1 + \epsilon)\text{dist}_X(q, s) + \delta \geq (1 + \epsilon) \inf_{x \in S_m} \text{dist}_X(q, x) + \delta = r_q.$$

For the reverse direction, express $\inf_{y \in S_m} \text{dist}_X(q, y)$ in terms of $r_q$:

$$\inf_{y \in S_m} \text{dist}_X(q, y) = \frac{r_q - \delta}{1 + \epsilon}.$$

By the triangle inequality, for any two points $y, s \in S_m$, we have

$$\text{dist}_X(s, q) \leq \text{dist}_X(y, q) + \text{dist}_X(s, y).$$

Since $\text{dist}_X(s, y) \leq \text{diam } S_m \leq (2 + \epsilon)\delta$, it follows that

$$\text{dist}_X(s, q) \leq \text{dist}_X(y, q) + (2 + \epsilon)\delta.$$

Combining this with the inequality $(1 + \epsilon)^2\text{dist}_X(y, q) \geq \text{dist}_X(y, q)$, and taking the infimum over $y \in S_m$, we obtain that for any point $s \in S_m$,

$$(1 + \epsilon)r_q + \delta = (1 + \epsilon)^2 \inf_{y \in S_m} \text{dist}_X(q, y) + (2 + \epsilon)\delta \geq \text{dist}_X(s, q).$$

Therefore, $r_q =_{\epsilon, \delta} \text{dist}_X(s, q)$ for every point $s \in S_m$, and hence the entire set $S_m$ lies within the feasible region. Once the reconstructor selects a guess point $\hat{x}$, the responder may choose any point from the feasible region, and in particular any $s \in S_m$.

By the definition of the Chebyshev radius,

$$r(S_m) \leq \sup_{x^* \in S_m} \text{dist}_X(x^*, \hat{x}),$$

so for any $\alpha > 0$, the responder can choose a point $x^* \in S_m$ such that $\text{dist}_X(\hat{x}, x^*) > r(S_m) - \alpha$.

It follows that

$$\text{OPT}(T, \epsilon, \delta) \geq r(S_m), \quad \text{and} \quad r(S_m) \xrightarrow[m \to \infty]{} e_X((2 + \epsilon)\delta).$$

Therefore,

$$\text{OPT}(T, \epsilon, \delta) \geq e_X((2 + \epsilon)\delta).$$

### C.2 Upper bound via $\alpha$-covers

To show that the optimal error is *at most* $e_X((2 + \epsilon)\delta)$, it suffices to construct a sequence of reconstructor strategies, each using $T_n$ queries, that guarantee a reconstruction error of at most $e_X((2 + \epsilon)\delta) + \alpha_n$, where $\alpha_n \to 0$.

Since the space $X$ is totally bounded, for any $\alpha > 0$ there exists a finite $\alpha$-cover $T_\alpha \subset X$, consisting of $T_\alpha$ points.

Take a sequence of $\alpha_n$-nets with $\alpha_n \to 0$, and denote the number of queries in the corresponding nets by $T_n := |T_{\alpha_n}|$. Since $X$ is totally bounded, these finite nets exist.

Denote the points of the $\alpha_n$-net by $\{q_t\}_{t \in [T_n]}$, and the responses of the responder by $\{r_t\}_{t \in [T_n]}$. We claim that

$$\text{diam}(\Phi(\{q_t, r_t\}_{t \in [T]})) \leq (2 + \epsilon)\delta + ((1 + \epsilon)^2 + 1)\alpha.$$

To see this, take *any* two points $A, B$ in the feasible region after the interaction.

There exists a query $q \in \{q_t\}_{t\in[T]}$ such that $\mathrm{dist}_X(A, q) \le \alpha_n$. Let $r$ be the responder's answer to this query. Since $A \in \Phi(q, r)$ and $\mathrm{dist}_X(q, A) \le \alpha_n$, we have

$$r \le (1 + \epsilon)\alpha_n + \delta.$$

On the other hand, since $B \in \Phi(q, r)$, we have

$$\mathrm{dist}_X(q, B) \le (1 + \epsilon)r + \delta \le (1 + \epsilon)^2 \alpha_n + (2 + \epsilon)\delta.$$

By the triangle inequality,

$$\mathrm{dist}_X(A, B) \le \mathrm{dist}_X(q, B) + \mathrm{dist}_X(A, q) \le \big((1 + \epsilon)^2 + 1\big)\alpha_n + (2 + \epsilon)\delta.$$

Hence $\mathrm{diam}\, \Phi(\{q_t, r_t\}_{t\in[T_n]}) \le (2 + \epsilon)\delta + ((1 + \epsilon)^2 + 1)\alpha_n$.

Denote by $\alpha_n'$ the quantity $((1+\epsilon)^2+1)\alpha_n$. The Chebyshev radius of $\Phi\big(\{(q_t, r_t)\}_{t\in[T_n]}\big)$ is therefore bounded by $\mathsf{e}_X\big((2 + \epsilon)\delta + \alpha_n'\big)$, and hence

$$\mathrm{OPT}_X(T_n, \epsilon, \delta) \le \mathsf{e}_X\big((2 + \epsilon)\delta + \alpha_n'\big).$$

By the right-continuity of $\mathsf{e}_X$ (see Appendix F.1), for every sequence of nonnegative numbers $\alpha_n' \to 0$ we have

$$\mathsf{e}_X\big((2 + \epsilon)\delta + \alpha_n'\big) \longrightarrow \mathsf{e}_X\big((2 + \epsilon)\delta\big).$$

Therefore, we conclude the desired bound

$$\mathrm{OPT}_X(\epsilon, \delta) \le \mathsf{e}_X\big((2 + \epsilon)\delta\big).$$

$\square$

The proof of the second part of the theorem relies on general properties of the function $\mathsf{e}_X$ that hold for arbitrary metric spaces.

**Theorem 2** (Second part). *If the distance $(2 + \epsilon)\delta$ is realized in a totally bounded metric space $X$, i.e., there exist a pair of points at this distance, then*

$$\frac{1}{2}(2 + \epsilon)\delta \le \mathrm{OPT}_X(\epsilon, \delta) \le (2 + \epsilon)\delta.$$

*Proof.* By the first part of Theorem 2, which we proved earlier,

$$\mathrm{OPT}_X(\epsilon, \delta) = \mathsf{e}_X((2 + \epsilon)\delta).$$

So it suffices to prove that

$$\frac{1}{2}(2 + \epsilon)\delta \le \mathsf{e}_X((2 + \epsilon)\delta) \le (2 + \epsilon)\delta.$$

To show the lower bound $\mathsf{e}_X((2 + \epsilon)\delta) \ge \frac{1}{2}(2 + \epsilon)\delta$, it suffices to construct a set $S \subseteq X$ of diameter at most $(2 + \epsilon)\delta$ such that *every* enclosing ball of $S$ must have radius at least $\frac{1}{2}(2 + \epsilon)\delta$. Indeed, by assumption, there exist two points $y_1, y_2 \in X$ such that

$$\mathrm{dist}_X(y_1, y_2) = (2 + \epsilon)\delta.$$

Then for any point $x \in X$, the triangle inequality implies

$$\mathrm{dist}_X(y_1, x) + \mathrm{dist}_X(x, y_2) \ge (2 + \epsilon)\delta,$$

so one of the two distances must be at least $\frac{1}{2}(2 + \epsilon)\delta$. Therefore, no point in $X$ lies at a distance less than $\frac{1}{2}(2 + \epsilon)\delta$ from both $y_1$ and $y_2$, and thus any ball containing both points must have radius at least this value. To show the upper bound $\mathsf{e}_X((2 + \epsilon)\delta) \le (2 + \epsilon)\delta$, it suffices to find an enclosing ball of radius $(2 + \epsilon)\delta$ for *any* set $S \subseteq X$ of diameter at most $(2 + \epsilon)\delta$. Indeed, let $x \in S$ be any point of the set, and consider the ball of radius $(2 + \epsilon)\delta$ centered at $x$. Since the diameter of $S$ is at most $(2 + \epsilon)\delta$, every point $y \in S$ satisfies $\mathrm{dist}_X(x, y) \le \mathrm{diam}(S) \le (2 + \epsilon)\delta$, so $S$ is entirely contained in this ball. This proves the claim. $\square$

# D   Feasible-region calculus

The goal of this section is to determine when there exists an answer $r \in \mathbb{R}^+$ such that a given set $S \subset X$ is contained in the feasible region $\Phi(q, r)$ (see Equation 2) resulting from a query at point $q$.

We will answer this question and provide a criterion for such an answer in Lemma 12.

For any set $S \subset X$, define its $\alpha$-neighborhood by

$$S_\alpha := \bigcup_{x \in S} B(x, \alpha),$$

where $B(x, \alpha)$ denotes the ball of radius $\alpha$ centered at $x$.

We will also be interested in the following optimization problem: what is the largest value of $\alpha$ such that the $\alpha$-neighborhood of a fixed set $S \subset X$ can be entirely contained in the feasible region for some answer to the query $q$? We will describe this quantity for convex Euclidean subspaces and specify the answer that the responder must give in order to preserve this neighborhood within the feasible region.

To answer the first question, it is useful to consider two natural candidates for the answer:

$$r_q^{\min}(S) := \frac{\sup_{s \in S} \operatorname{dist}_X(s, q) - \delta}{1 + \epsilon}, \qquad r_q^{\max}(S) := (1 + \epsilon) \inf_{s \in S} \operatorname{dist}_X(s, q) + \delta.$$

Intuitively, $r_q^{\min}(S)$ places the farthest point of $S$ on the outer boundary of the feasible region while keeping all of $S$ inside it, and $r_q^{\max}(S)$ places the nearest point of $S$ on the inner boundary while still preserving inclusion (see Fig. 3, Fig. 4).

Supremum and infimum of the set of numbers $\{\operatorname{dist}_X(y, s)\}_{s \in S}$ play the same role as $\min_{s \in S} \operatorname{dist}_X(s, q)$ and $\max_{s \in S} \operatorname{dist}_X(s, q)$. The only difference is that *sometimes* the minimum or maximum is not attained by any point in the set $S$. In such cases, one must take a sequence of points $\{x_i\}_{i \in \mathbb{N}} \subseteq S$ that plays the role of the minimum or maximum, in the sense that

$$\lim_{i \to \infty} \operatorname{dist}_X(y, x_i) = \sup_{s \in S} \operatorname{dist}_X(y, s) \quad \text{or} \quad \lim_{i \to \infty} \operatorname{dist}_X(y, x_i) = \inf_{s \in S} \operatorname{dist}_X(y, s).$$

**Lemma 12** (Consistency window). *Fix a set $S \subset X$. For a given query $q$, there exists an answer $r$ such that $S$ is contained in the feasible region $\Phi(q, r)$ if and only if $r_q^{\min}(S) \leq r_q^{\max}(S)$. Moreover, this inclusion holds if and only if $r \in [r_q^{\min}(S), r_q^{\max}(S)]$[8].*

*Proof.* Assume the responder gives an answer $r$ such that $S \subset \Phi(q, r)$. By the definition of the supremum, if

$$r < \frac{\sup_{x \in S} \operatorname{dist}_X(x, q) - \delta}{1 + \epsilon},$$

then there exists a point $A \in S$ such that

$$r < \frac{\operatorname{dist}_X(A, q) - \delta}{1 + \epsilon},$$

and hence $A \notin \Phi(q, r)$, contradicting the assumption.

Similarly, if $r > r_q^{\max}(S) = (1 + \epsilon) \inf_{s \in S} \operatorname{dist}_X(s, q) + \delta$, then there exists a point $B \in S$ such that

$$r > (1 + \epsilon) \operatorname{dist}_X(B, q) + \delta,$$

and therefore $B \notin \Phi(q, r)$.

Hence, for any $r$ outside the interval $[r_q^{\min}(S), r_q^{\max}(S)]$, there exists a point in $S$ that lies outside $\Phi(q, r)$. This shows that if $S \subset \Phi(q, r)$, then necessarily $r \in [r_q^{\min}(S), r_q^{\max}(S)]$, and in particular $r_q^{\min}(S) \leq r_q^{\max}(S)$.

---

[8]Note that we do not require the answer to be positive; it may be negative, yet the feasible region can still be non-empty.

To prove the converse, suppose $r \in [r_q^{\min}(S), r_q^{\max}(S)]$. Take any point $s \in S$. We must verify the two inequalities:

$$\text{dist}_X(s,q) \leq (1+\epsilon)r + \delta \quad \text{and} \quad r \leq (1+\epsilon)\text{dist}_X(s,q) + \delta.$$

Indeed, since

$$r \leq r_q^{\max}(S) = (1+\epsilon)\inf_{x \in S}\text{dist}_X(x,q) + \delta \leq (1+\epsilon)\text{dist}_X(s,q) + \delta,$$

and

$$r \geq r_q^{\min}(S) = \frac{\sup_{x \in S}\text{dist}_X(x,q) - \delta}{1+\epsilon} \geq \frac{\text{dist}_X(s,q) - \delta}{1+\epsilon},$$

we have $s \in \Phi(q,r)$. Since $s \in S$ was arbitrary, it follows that $S \subset \Phi(q,r)$, completing the proof. $\qquad \square$

The observation above does not rely on any structural properties of the metric space; in particular, it holds even if the triangle inequality is not satisfied. However, to determine the largest neighborhood of a set that may remain feasible after a query, we need to use some form of continuity in the space. That's why, from this point on, we assume that the metric space $X$ is a convex subset of $\mathbb{R}^n$.

**Observation 13.** *Let $S \subseteq X$, and let $\alpha > 0$. Suppose that for every $x \in S$, the Euclidean ball $B(x,\alpha) \subseteq \mathbb{R}^n$ is entirely contained in $X$. Then:*

$$r_q^{\min}(S_\alpha) = r_q^{\min}(S) + \frac{\alpha}{1+\epsilon}, \qquad r_q^{\max}(S_\alpha) = \max\left\{\delta,\, r_q^{\max}(S) - \alpha(1+\epsilon)\right\}.$$

*Proof.* We start by analyzing the supremum. We want to show:

$$\sup_{y \in S_\alpha}\text{dist}_X(q,y) = \sup_{s \in S}\text{dist}_X(s,q) + \alpha.$$

The inequality

$$\sup_{y \in S_\alpha}\text{dist}_X(q,y) \leq \sup_{s \in S}\text{dist}_X(s,q) + \alpha$$

is immediate from the triangle inequality. Indeed, for any point $y \in S_\alpha$, there exists some $s \in S$ such that $y \in B(s,\alpha)$. Then:

$$\text{dist}_X(q,y) \leq \text{dist}_X(q,s) + \text{dist}_X(s,y) \leq \sup_{s \in S}\text{dist}_X(s,q) + \alpha,$$

and so the inequality holds for all $y \in S_\alpha$, yielding the upper bound on the supremum.

For the reverse inequality, take a sequence $\{x_i\}_{i \in \mathbb{N}} \subset S$ such that:

$$\lim_{i \to \infty}\text{dist}_X(x_i,q) = \sup_{s \in S}\text{dist}_X(s,q).$$

For each $x_i$, choose a point $y_i \in B(x_i,\alpha)$ lying along the ray from $q$ through $x_i$ such that:

$$\text{dist}_X(q,y_i) = \text{dist}_X(q,x_i) + \alpha.$$

This is possible because the balls are Euclidean. Then:

$$\sup_{y \in S_\alpha}\text{dist}_X(q,y) \geq \lim_{i \to \infty}\text{dist}_X(q,y_i) = \sup_{s \in S}\text{dist}_X(s,q) + \alpha,$$

proving the desired equality.

Now we turn to the infimum:

$$\inf_{y \in S_\alpha}\text{dist}_X(q,y) = \max\left\{0,\, \inf_{s \in S}\text{dist}_X(s,q) - \alpha\right\}.$$

First, by the triangle inequality again, we have:

$$\text{dist}_X(q,y) \geq \text{dist}_X(q,s) - \text{dist}_X(y,s) \geq \inf_{s \in S}\text{dist}_X(s,q) - \alpha$$

for all $y \in S_\alpha$. Also, clearly $\mathrm{dist}_X(q, y) \geq 0$. Therefore,

$$\inf_{y \in S_\alpha} \mathrm{dist}_X(q, y) \geq \max\left\{0, \inf_{s \in S} \mathrm{dist}_X(s, q) - \alpha\right\}.$$

To show the reverse inequality, consider a sequence $\{x_i\}_{i \in \mathbb{N}} \subset S$ such that:

$$\lim_{i \to \infty} \mathrm{dist}_X(x_i, q) = \inf_{s \in S} \mathrm{dist}_X(s, q).$$

If $\inf_{s \in S} \mathrm{dist}_X(s, q) < \alpha$, then for some $x_j$, we have $\mathrm{dist}_X(x_j, q) < \alpha$, so $q \in B(x_j, \alpha) \subset S_\alpha$, and thus:

$$\inf_{y \in S_\alpha} \mathrm{dist}_X(q, y) = 0.$$

Otherwise, all $x_i$ satisfy $\mathrm{dist}_X(x_i, q) \geq \alpha$. In that case, for each $x_i$, there exists a point $y_i \in [q, x_i]$ such that $\mathrm{dist}_X(x_i, y_i) = \alpha$, i.e., $y_i$ lies along the segment from $q$ to $x_i$, at distance $\alpha$ from $x_i$. Then:

$$\mathrm{dist}_X(y_i, q) = \mathrm{dist}_X(x_i, q) - \alpha,$$

and so:

$$\inf_{y \in S_\alpha} \mathrm{dist}_X(q, y) \leq \lim_{i \to \infty} \mathrm{dist}_X(y_i, q) = \inf_{s \in S} \mathrm{dist}_X(s, q) - \alpha.$$

This completes the proof. $\qquad\square$

Let us denote

$$\rho_{\min}^q(S) := \inf_{s \in S} \mathrm{dist}_X(q, s), \qquad \rho_{\max}^q(S) := \sup_{s \in S} \mathrm{dist}_X(q, s).$$

These quantities represent the minimal and maximal distances from the query point $q$ to the set $S$, and will be used to simplify the expressions that follow.

**Lemma 14.** *Fix a set $S \subset X$ and a query point $q \in X$. Define*

$$\alpha^\star = \frac{r_q^{\max}(S) - r_q^{\min}(S)}{(1 + \epsilon) + \frac{1}{1+\epsilon}}.$$

*Assume that $\alpha^\star > 0$, and that for every point $s \in S$, the Euclidean ball $B(s, \alpha^\star)$, viewed as a subset of $\mathbb{R}^n$, is contained in $X$; that is, $B(s, \alpha^\star) \subseteq X$.*

*Then there exists an answer $r$ such that $S_{\alpha^\star} \subset \Phi(q, r)$. In particular, for the specific choice*

$$r_q^\star(S) := \frac{(1 + \epsilon)\left(\rho_{\min}^q + \rho_{\max}^q\right) - \epsilon\delta}{1 + (1 + \epsilon)^2},$$

*we have $S_{\alpha^\star} \subset \Phi(q, r_q^\star(S))$.*

*Proof.* By Lemma 12, it suffices to verify that

$$r_q^{\min}(S_{\alpha^\star}) \leq r_q^\star(S) \leq r_q^{\max}(S_{\alpha^\star}).$$

First, observe that

$$\rho_{\max}^q = (1 + \epsilon)r_q^{\min}(S) + \delta, \qquad \rho_{\min}^q = \frac{r_q^{\max}(S) - \delta}{1 + \epsilon}$$

and therefore

$$r_q^\star(S) = \frac{r_q^{\max} + (1 + \epsilon)^2 r_q^{\min}}{(1 + \epsilon)^2 + 1}.$$

To verify the lower bound, note that by Observation 13,

$$
\begin{aligned}
r_q^{\min}(S_{\alpha^\star}) &= r_q^{\min}(S) + \frac{\alpha^\star}{1+\epsilon} \\
&= r_q^{\min}(S) + \frac{r_q^{\max}(S) - r_q^{\min}(S)}{(1+\epsilon)^2 + 1} \\
&= \frac{(1+\epsilon)^2 \cdot r_q^{\min}(S) + r_q^{\max}(S)}{(1+\epsilon)^2 + 1} \\
&= r_q^\star(S).
\end{aligned}
$$

For the upper bound, we distinguish between two cases depending on whether $\rho_{\min}^q \geq \alpha^\star$ or $\rho_{\min}^q < \alpha^\star$. Indeed, the form of $r_q^{\max}(S_{\alpha^\star})$ is case-dependent, with two distinct formulas: if $\rho_{\min}^q \geq \alpha^\star$, then $r_q^{\max}(S_{\alpha^\star}) = r_q^{\max}(S) - (1+\epsilon)\alpha^\star$; otherwise, $r_q^{\max}(S_{\alpha^\star}) = \delta$.

**Case** $\rho_{\min}^q \geq \alpha^\star$. Then by Observation 13,

$$
\begin{aligned}
r_q^{\max}(S_{\alpha^\star}) &= r_q^{\max}(S) - (1+\epsilon)\alpha^\star \\
&= r_q^{\max}(S) - \frac{(1+\epsilon)^2 \left( r_q^{\max}(S) - r_q^{\min}(S) \right)}{(1+\epsilon)^2 + 1} \\
&= \frac{(1+\epsilon)^2 \cdot r_q^{\min}(S) + r_q^{\max}(S)}{(1+\epsilon)^2 + 1} \\
&= r_q^\star(S).
\end{aligned}
$$

**Case** $\rho_{\min}^q < \alpha^\star$. In this case, we observe that

$$
r_q^{\max}(S) = (1+\epsilon) \cdot \rho_{\min}^q + \delta < (1+\epsilon)\alpha^\star + \delta,
$$

and therefore

$$
\delta > r_q^{\max}(S) - (1+\epsilon)\alpha^\star = r_q^\star(S),
$$

so again $r_q^\star(S) < \delta = r_q^{\max}(S_{\alpha^\star})$, as required.

This completes the proof. $\qquad\square$

For later use, we express the quantity $\alpha^\star$ in terms of $\rho_{\min}^q$ and $\rho_{\max}^q$, since this representation will be useful below:

$$
\alpha^\star = \frac{(1+\epsilon)^2 \rho_{\min}^q - \rho_{\max}^q + (2+\epsilon)\delta}{(1+\epsilon)^2 + 1}.
$$

This leads to the following observation.

**Remark 15.** *Assume that* $\operatorname{diam} S \leq (2+\epsilon)\delta$. *Then the radius* $\alpha^\star$ *of the neighborhood* $S_{\alpha^\star}$, *as defined in Lemma 14, can be decomposed into two nonnegative terms,* $\alpha^\star = \alpha_1 + \alpha_2$, *where*

$$
\alpha_1 = \frac{(2+\epsilon)\delta - \left( \sup_{x \in S} \operatorname{dist}_X(x,q) - \inf_{x \in S} \operatorname{dist}_X(x,q) \right)}{(1+\epsilon)^2 + 1}
$$

*and*

$$
\alpha_2 = \frac{(1+\epsilon)^2 - 1}{(1+\epsilon)^2 + 1} \cdot \inf_{x \in S} \operatorname{dist}_X(x,q).
$$

*Note that* $\alpha_1 \geq 0$, *since* $\sup_{x \in S} \operatorname{dist}_X(x,q) - \inf_{x \in S} \operatorname{dist}_X(x,q) \leq \operatorname{diam} S \leq (2+\epsilon)\delta$.

**Observation 16.** *Assume* $\epsilon = 0$ *and fix the regular simplex* $\Delta \subset X$ *with edges of length* $2\delta$, *and the query* $q \in \mathbb{R}^n$. *Denote*

$$
A = \operatorname{argmax}_{A_i \in \Delta} \operatorname{dist}_X(A,q), \qquad B = \operatorname{argmin}_{A_i \in \Delta} \operatorname{dist}_X(A,q).
$$

*Then Lemma 14 can be simplified; the* $\alpha^\star$*-neighborhood lies in the feasible region:* $\Delta_{\alpha^\star} \subset \Phi(q, r_q^\star(\Delta))$ *for*

$$
r_q^\star(\Delta) = \frac{\operatorname{dist}_X(q,A) + \operatorname{dist}_X(q,B)}{2},
$$

$$
\alpha^\star = \frac{\operatorname{dist}_X(q,A) - \operatorname{dist}_X(q,B)}{2}.
$$

**Lemma 17.** *Assume we are in the $\epsilon = 0$ game scenario. Fix $\delta > \alpha > 0$, and let $\Delta = A_0 A_1 \ldots A_n$ be a regular simplex with edges of length $2\delta$. For a given point $q$, let $A \in \Delta$ be the farthest vertex from $q$, and let $B \in \Delta$ be the nearest vertex (in the case of ties, choose any).*

*If*

$$\cos \angle BAq \;\le\; 1 - \frac{\alpha}{\delta},$$

*then*

$$\Delta_\alpha \;\subset\; \Phi\big(\{q, r_q^\star(\Delta)\}\big), \qquad r_q^\star(\Delta) = \frac{qA + qB}{2}.$$

*Proof.* It suffices to show that $B(A, \alpha)$ and $B(B, \alpha)$ are both contained in $\Phi(q, r_q^\star(\Delta))$.

Let us estimate $Aq - Bq$. We will show that

$$Aq - Bq \le \cos \angle BAq \cdot 2\delta.$$

Drop a perpendicular from $q$ onto the line $AB$, and let the foot of this perpendicular be $H$. Denote the angle $\angle qAH$ by $\phi$ and the angle $\angle qBH$ by $\psi$.

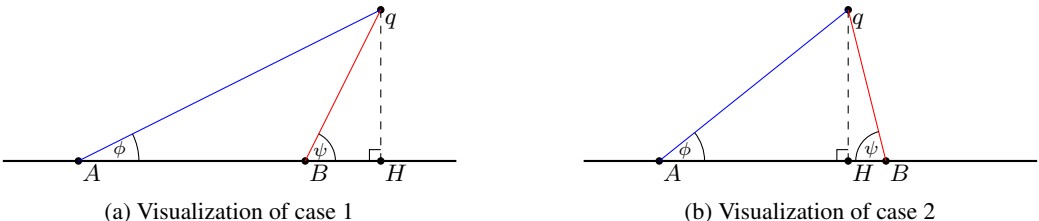

(a) Visualization of case 1          (b) Visualization of case 2

Figure 5: Visualization of two cases

We distinguish two cases: $H \notin AB$ and $H \in AB$ (see Figure 5). Note that in both cases,

$$0 < \phi \le \psi < \frac{\pi}{2},$$

and since the cosine function decreases on this interval, we have $\cos \phi \ge \cos \psi \ge 0$.

**Case 1:** $H \notin AB$. Since $Aq^2 = qH^2 + HA^2$ and $Bq^2 = BH^2 + HB^2$, and $AH - BH = 2\delta$ we compute:

$$Aq - Bq \;=\; \frac{Aq^2 - Bq^2}{Aq + Bq} \;=\; \frac{AH^2 - BH^2}{Aq + Bq} \;=\; 2\delta \, \frac{AH + BH}{Aq + Bq}.$$

On the other hand since $\cos \phi \ge \cos \psi$:

$$\frac{AH + BH}{Aq + Bq} = \frac{Aq \cdot \cos \phi + Bq \cdot \cos \psi}{Aq + Bq} \le 1.$$

Hence

$$Aq - Bq \;\le\; 2\delta \cos \phi = 2\delta \cos \angle qAB.$$

**Case 2:** $H \in AB$. Similarly since $AH + BH = 2\delta$:

$$Aq - Bq \;=\; \frac{Aq^2 - Bq^2}{Aq + Bq} \;=\; \frac{AH^2 - BH^2}{Aq + Bq} \;=\; 2\delta \, \frac{AH - BH}{Aq + Bq} \;=\;$$
$$2\delta \, \frac{Aq \cdot \cos \phi - Bq \cdot \cos \psi}{Aq + Bq},$$

which again implies

$$Aq - Bq \;\le\; 2\delta \cos \phi = 2\delta \cos \angle qAB.$$

Now take a point $M_A \in B(A, \alpha)$. By Lemma assumption $\alpha \le \delta(1 - \cos \angle BAq)$, hence

$$qM_A \;\le\; qA + \alpha \;\le\; qA + \delta(1 - \cos \angle BAq) \;\le\; qA - \frac{Aq - Bq}{2} + \delta \;=\; r_q^\star(\Delta) + \delta.$$

Also, because $\alpha < \delta$, it follows that $M_A A \geq r_q^\star(\Delta)$. Hence $M_A \in \Phi(q, r_q^\star(\Delta))$.

A similar argument applies to any $M_B \in B(B, \alpha)$. Indeed,

$$qM_B \;\geq\; qB - \alpha \;\geq\; qB - \delta(1 - \cos\angle BAq) \;\geq\; r_q^\star(\Delta) - \delta,$$

and again $\alpha < \delta$ implies $M_B \in \Phi(q, r_q^\star(\Delta))$. Thus $B(A, \alpha)$ and $B(B, \alpha)$ lie in $\Phi(q, r_q^\star(\Delta))$, completing the proof. $\qquad\square$

## E   Proof of Theorem 6

**Theorem** (Theorem 6 Restatement). *Let $X \subset \mathbb{R}^n$ be a bounded convex set equipped with the Euclidean metric such that $\dim X > 0$, and let $\epsilon \geq 0$. Then, for all sufficiently small $\delta > 0$, the space $X$ is* not *$(\epsilon, \delta)$-pseudo-finite, except in the case where $\epsilon = 0$ and $\dim X = 1$.*

**Proof.**   Since the reconstruction game is played entirely within the space $X$—in the sense that all queries, the final guess, and the secret point lie in $X$—we may assume without loss of generality that $X \subset \mathbb{R}^n$, where $n = \dim(X)$ is the affine dimension of $X$.

We begin with the special and simple case where $\epsilon = 0$ and $\dim X = 1$. In this case, $X$ is an interval with endpoints $a, b$; without loss of generality, we assume that both $a, b \in X$. (The cases where $X$ is half-open or open can be handled similarly.) If $\delta \geq (b - a)/2$, then the optimal error equals the diameter of the space, and the reconstructor can trivially achieve it by outputting any point in $X$ without submitting any queries. Otherwise, if $\delta < (b - a)/2$, the reconstructor submits a single query at one of the endpoints, say $q_1 = a$, and receives a $r_1$). The feasible region then becomes an interval of length at most $2\delta$, and by guessing its midpoint, the reconstructor achieves the optimal approximation error of $\delta$.

The remaining cases—when either $\epsilon > 0$ or $\dim X \geq 2$—are more challenging and constitute the core of the proof. We divide the proof into two cases: one where $\epsilon = 0$, and one where $\epsilon > 0$. In both cases, the proof follows a similar strategy. We show that for all sufficiently small $\delta > 0$, there exists a responder strategy that guarantees, for every number of rounds $T$, that the feasible region contains an extremal simplex $\Delta_T$ whose Chebyshev radius is strictly greater than the optimal value $\mathrm{OPT}(\epsilon, \delta)$. This suffices to prove that the optimal error cannot be attained in finite time.

More precisely, we show that for each $t = 0, 1, \ldots, T$, the responder can ensure that the feasible region contains an $\alpha_t$-neighborhood of a regular simplex $\Delta_t$ of diameter exactly $(2 + \epsilon)\delta$, where the neighborhood is defined as the union of all balls of radius $\alpha_t$ centered at the vertices of $\Delta_t$. Since such a neighborhood contains a regular simplex of diameter $(2 + \epsilon)\delta + \sqrt{\frac{2(n+1)}{n}}\alpha_t$, it follows that the Chebyshev radius of the feasible region is strictly greater than $\mathrm{OPT}(\epsilon, \delta)$, which corresponds to the Chebyshev radius of a regular simplex of diameter $(2 + \epsilon)\delta$.

The proof proceeds inductively. We assume that after $t$ rounds, the feasible region contains an $\alpha_t$-neighborhood of a regular simplex $\Delta_t$ of diameter $(2+\epsilon)\delta$, and we show that for any query $q_{t+1} \in X$, there exists a response such that the updated feasible region contains an $\alpha_{t+1}$-neighborhood of some (possibly different) regular simplex $\Delta_{t+1}$ of the same diameter. Moreover, $\alpha_{t+1} = f_{\epsilon,\delta}(\alpha_t)$, where the function $f_{\epsilon,\delta}$ is defined by:

$$f_{\epsilon,\delta}(\alpha) = \begin{cases} \frac{\alpha^2}{2 \cdot 81\,\delta}, & \epsilon = 0, \\ \frac{(1+\epsilon)^2 - 1}{2(1+\epsilon)^2} \cdot \alpha, & \epsilon > 0. \end{cases}$$

Thus, the responder can recursively maintain an $\alpha_t$-neighborhood of a regular simplex throughout the game, where $\alpha_t = f_{\epsilon,\delta}^{(t)}(\alpha_0)$, and $\alpha_0$ is the maximum value such that $X$ contains an $\alpha_0$-neighborhood of a regular simplex of diameter $(2 + \epsilon)\delta$. This completes the high-level argument. To complete the inductive proof, it remains to establish the base case and then develop the tools needed for the inductive step.

**Base Case.**   Since $X$ is a bounded convex subset of $\mathbb{R}^n$ with nonempty interior, it follows that for any $\epsilon > 0$, there exists a sufficiently small $\delta > 0$ and some $\alpha_0 > 0$ such that $X$ contains an $\alpha_0$-neighborhood of a regular simplex $\Delta^0$ with diameter $(2 + \epsilon)\delta$. This establishes the base case for

our induction: at round $t = 0$, the feasible region contains a neighborhood $\Delta^0_{\alpha_0} \subseteq X$ that satisfies the required conditions.

**Inductive Step.** The remainder of the proof develops the geometric tools needed to carry out the inductive step, namely to show that such a neighborhood can be maintained (with decreasing radius) after each query. We now introduce the some notation used throughout the argument.

**Notation 18.** *Let $0 < \alpha \leq \delta$, and let $\Delta = \{A_0, A_1, \ldots, A_n\}$ be a regular simplex in $\mathbb{R}^n$ of diameter $(2 + \epsilon)\delta$. We define the $\alpha$-neighborhood of $\Delta$, denoted $\Delta_\alpha$, as the union of closed Euclidean balls of radius $\alpha$ centered at the vertices of $\Delta$.*

*A query point $q \in \mathbb{R}^n$ is called* good *(with respect to $\Delta$ and $\alpha$) if there exists a response $r$ such that the entire neighborhood $\Delta_\alpha$ is contained in the feasible region $\Phi(q, r)$; otherwise, $q$ is called* bad.

*We will use the special response $r = r_q^\star(\Delta)$ defined in Lemma 14 to ensure that the neighborhood $\Delta_{\alpha_{t+1}}$ remains feasible; this choice will be sufficient for our inductive argument.*

### E.1 Case $\epsilon > 0$ (translation strategy)

Assume the responder receives the query $q$, and has so far managed to keep the $\alpha$-neighborhood of a regular simplex $\Delta \subset \mathbb{R}^n$, with edge length $(2 + \epsilon)\delta$, inside the feasible region. Without loss of generality, we may assume that $4\alpha \leq (2 + \epsilon)\delta$.

In this subsection, we will show that when $\epsilon > 0$, there exists another regular simplex $\Delta'$ with the same edge length such that

$$\Delta'_{\alpha'} \subset \Delta_\alpha, \qquad \Delta'_{\alpha'} \subset \Phi(q, r_q^\star(\Delta')), \quad \text{where} \quad \alpha' = \frac{(1+\epsilon)^2 - 1}{2(1+\epsilon)^2}\alpha.$$

This will be sufficient to establish the induction step for the case $\epsilon > 0$.

In the terminology of Notation 18, the query point $q$ may be either good or bad with respect to the simplex $\Delta$ and neighborhood $\alpha'$. If $q$ is good, we are done by simply taking $\Delta' := \Delta$.

If the point $q$ is bad, we will use the decomposition of the neighborhood sustained by the answer $r_q^\star(\Delta)$, as described in Remark 15, in order to construct a new simplex.

Let us remind the reader that, by Lemma 15, there is a formula for the radius $\alpha^\star$ of the neighborhood $\Delta_{\alpha^\star}$, which corresponds to the feasible region after answering with $r_q^\star(\Delta)$. Remark 15 states that this radius can be decomposed into two nonnegative terms.

In the case $\epsilon > 0$, the second term will be of particular interest:

$$\alpha^\star = \alpha_1 + \alpha_2, \qquad \alpha_2 = \frac{(1+\epsilon)^2 - 1}{(1+\epsilon)^2 + 1} \cdot \inf_{x \in \Delta} \text{dist}_X(x, q).$$

Since we assumed that the query $q$ is bad with respect to the simplex $\Delta$ and neighborhood $\alpha'$, it follows that

$$\alpha' > \frac{(1+\epsilon)^2 - 1}{(1+\epsilon)^2 + 1} \cdot \inf_{x \in \Delta} \text{dist}_X(x, q).$$

In particular, this implies

$$\text{dist}_X(B, q) < \alpha' \cdot \frac{(1+\epsilon)^2 + 1}{(1+\epsilon)^2 - 1} = \frac{(1+\epsilon)^2 + 1}{2(1+\epsilon)^2} \cdot \alpha = \alpha - \alpha',$$

where $B := \arg\min_{x \in \Delta} \text{dist}_X(x, q)$ is the closest vertex of $\Delta$ to the query point $q$.

Define the shifted simplex $\Delta' := \Delta + \vec{v}$, where the vector $\vec{v}$ is in the same direction as the vector $\overrightarrow{qB}$, and its length is

$$\|\vec{v}\| = \alpha - \alpha' = \frac{2(1+\epsilon)^2}{(1+\epsilon)^2 - 1}\alpha' - \alpha' = \frac{(1+\epsilon)^2 + 1}{(1+\epsilon)^2 - 1} \cdot \alpha'.$$

In the degenerate case when $q = B$, choose any vector $\vec{v}$ of that length.

The shifted neighborhood satisfies $\Delta'_{\alpha'} \subset \Delta_\alpha$. Assume $x \in \Delta'_{\alpha'}$; then there exists a shifted vertex $A' := A + \vec{v}$ (that is, $A'$ is the image of $A$ under translation by the vector $\vec{v}$) such that $x \in B(A', \alpha')$.

Let us denote by $B'$ the point obtained by shifting the vertex $B$ of $\Delta$ (the one closest to $q$) by the vector $\vec{v}$.

We claim that $B'$ is the nearest vertex of the translated simplex $\Delta'$ to the query point $q$. This follows from the triangle inequality and the bounds

$$\mathrm{dist}_X(B, q) \leq \alpha - \alpha', \qquad \alpha < \frac{(2 + \epsilon)\delta}{4}$$

Indeed, consider any other vertex $C' = C + \vec{v}$ of $\Delta'$, where $C \neq B$. Then:

$$\mathrm{dist}_X(q, C') \geq \mathrm{dist}_X(q, C) - \|\vec{v}\| = \mathrm{dist}_X(q, C) - (\alpha - \alpha').$$

On the other hand, we have

$$\mathrm{dist}_X(q, C) \geq \mathrm{dist}_X(C, B) - \mathrm{dist}_X(q, B) \geq (2 + \epsilon)\delta - (\alpha - \alpha').$$

Combining these inequalities gives

$$\mathrm{dist}_X(q, C') \geq (2 + \epsilon)\delta - 2(\alpha - \alpha').$$

Meanwhile, the distance from $q$ to $B'$ satisfies

$$\mathrm{dist}_X(q, B') \leq \mathrm{dist}_X(q, B) + (\alpha - \alpha') \leq 2(\alpha - \alpha').$$

Since we assumed $\alpha < \frac{(2+\epsilon)\delta}{4}$, it follows that

$$\mathrm{dist}_X(q, B') \leq 2(\alpha - \alpha') < (2 + \epsilon)\delta - 2(\alpha - \alpha') \leq \mathrm{dist}_X(q, C'),$$

which confirms that $B'$ is indeed the closest vertex of $\Delta'$ to $q$.

Since $B + \vec{v}$ is the vertex of the new simplex $\Delta'$ closest to $q$, we have

$$\mathrm{dist}_X(q, B + \vec{v}) \geq \|\vec{v}\| = \frac{(1 + \epsilon)^2 + 1}{(1 + \epsilon)^2 - 1} \cdot \alpha'.$$

Therefore, by Remark 15 (which follows from Lemma 14), the entire neighborhood $\Delta'_{\alpha'}$ is contained in the feasible region:

$$\Delta'_{\alpha'} \subset \Phi(q, r_q^\star(\Delta')).$$

This completes the argument.

### E.2 Case $\epsilon = 0$ (rotation strategy)

The strategy for handling the case $\epsilon = 0$ will be similar. Assume the responder receives a query $q$, and that the $\alpha$-neighborhood of a regular simplex $\Delta$, with edge length $2\delta$, is contained in the feasible region. Without loss of generality we may assume that $\alpha < \frac{\delta}{4}$.

We will again show that there exists another regular simplex $\Delta'$, with the same edge length, such that

$$\Delta'_{\alpha'} \subset \Delta_\alpha, \qquad \Delta'_{\alpha'} \subset \Phi(q, r_q^\star(\Delta')), \quad \text{where} \quad \alpha' = \frac{\alpha^2}{81 \cdot 2\delta}.$$

In the case $\epsilon > 0$, the $\alpha'$-good points were those whose distances to the simplex $\Delta$ were sufficiently big. Obviously, when $\epsilon = 0$, this method of locating good points no longer works: for example, the entire line passing through two vertices $A$ and $B$ of the simplex $\Delta$ consists of $\alpha$-bad points for any $\alpha > 0$.

Note also that in our earlier argument—where we moved the simplex so that a previously bad point would become good with respect to the shifted simplex—we did not require a full characterization of bad points. It was enough to identify a property shared by all bad points and then move the simplex so that the given point no longer satisfies that property.

The same strategy applies in the case $\epsilon = 0$, using Lemma 17. Let $B := \arg\min_{A_i \in \Delta} \mathrm{dist}_X(q, A_i)$ be the nearest vertex of $\Delta$ to the query point $q$, and let $A := \arg\max_{A_i \in \Delta} \mathrm{dist}_X(q, A_i)$ be the farthest.

By Lemma 17, if the point $q$ is bad, then the angle between the vectors $\overrightarrow{AB}$ and $\overrightarrow{Aq}$—that is, the angle $\angle qAB$—must be sufficiently small. Why is this the case?

Note that since $B$ is closer to $q$ than $A$, the angle $\angle qAB$ lies in the interval $[0, \pi/2)$. Over this interval, the cosine function decreases from 1 to 0. Therefore, if the point $q$ is $\alpha'$-bad, we must have

$$\cos(\overrightarrow{AB}, \overrightarrow{Aq}) > 1 - \frac{\alpha'}{\delta} \quad \implies \quad \angle BAq < \arccos\left(1 - \frac{\alpha'}{\delta}\right).$$

The regions consisting of points satisfying $\angle BAq < \arccos\left(1 - \frac{\alpha'}{\delta}\right)$ are illustrated in Figure 6.

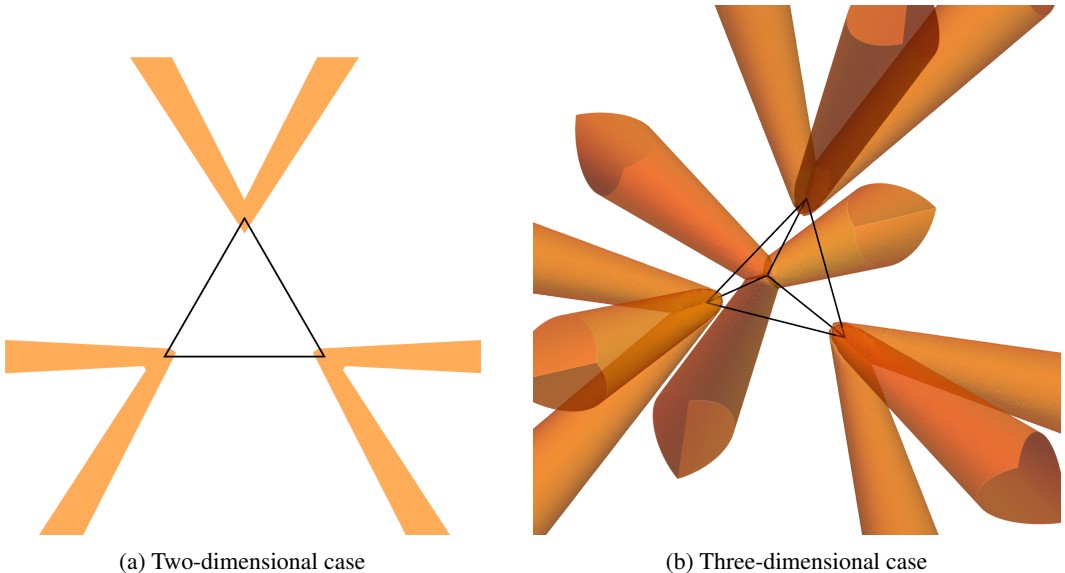

(a) Two-dimensional case $\qquad\qquad\qquad$ (b) Three-dimensional case

Figure 6: All bad points lie in the orange region

The proof proceeds as follows. We assume that the point $q$ satisfies

$$\angle BAq < \arccos\left(1 - \frac{\alpha'}{\delta}\right).$$

Otherwise, by the argument above, the point $q$ is $\alpha'$-good, and we can provide the answer using Lemma 17.

We then construct a isometry $\gamma$ such that the transformed point $q' := \gamma(q)$ no longer satisfies this property. That is, let $A'$ and $B'$ be the farthest and nearest[9] vertices of $\Delta$ with respect to $q'$. Then the angle $\angle B'A'q'$ satisfies

$$\angle B'A'q' \geq \arccos\left(1 - \frac{\alpha'}{\delta}\right).$$

Notice that it would be sufficient to construct such an isometry: if the point $\gamma(q)$ is $\alpha'$-good with respect to the simplex $\Delta$, then the original point $q$ is $\alpha'$-good with respect to the transformed simplex $\Delta' := \gamma^{-1}(\Delta)$. Once such a rotation is constructed, the remaining task is to argue that $\Delta'_{\alpha'} \subset \Delta_{\alpha}$.

Denote by $a := \frac{\alpha}{2\delta}$ and $b := \frac{\alpha'}{2\delta}$ the normalized neighborhood radii.

The main challenge in constructing such an isometry is that, if we are not careful—say, we transform the space in a way that ensures $\angle BAq' \geq \arccos\left(1 - \frac{\alpha'}{\delta}\right)$—the point $q'$ may still turn out to be bad. This can happen because the isometry might unintentionally change the identity of the farthest or nearest vertex of the simplex with respect to $q'$.

---

[9]We will even prove that the transformation preserves the identities of the nearest and farthest vertices.

This is why in Lemma 23 we constructed a small enough isometry $R_{2\theta}$ for the specific query $q$ such that it does not change the identity of the farthest or nearest vertex of the simplex with respect to $R_{2\theta}[q]$.

Define $\theta := \arccos(1 - 2b)$, and consider the rotation $R_{2\theta}$, defined at the beginning of Appendix F.2 (see Equation 4), associated with the query $q$ and the simplex $\Delta$. Recall that this rotation acts on the plane $\Pi := \mathrm{span}(A, B, Q)$—where $Q$ is the centroid of the remaining vertices in $\Delta \setminus \{A, B\}$—by rotating around the point $A$ through an angle of $2\theta$. On the orthogonal complement $\Pi^\perp$, it acts as the identity: $R_{2\theta}\big|_{\Pi^\perp} = \mathrm{Id}_{\Pi^\perp}$.

Since we assumed that the point $q$ is $\alpha'$-bad, we must have

$$\angle BAq < \arccos(1 - 2b) = \theta.$$

Notice also that, since we assumed $\alpha < \frac{\delta}{4}$, it follows that $\alpha' < \frac{\delta}{32 \cdot 81}$. Therefore,

$$1 - 2 \cdot \frac{\alpha'}{2\delta} = 1 - \frac{1}{32 \cdot 81} > \cos\left(\frac{\pi}{18}\right),$$

which ensures that $\theta < \frac{\pi}{18}$. This inequality can be verified using standard analysis tools, such as Taylor expansions of the cosine function.

Hence, we may apply Lemma 23, which states that the farthest and nearest vertices with respect to $q' := R_{2\theta}(q)$ are preserved under this isometry whenever $\theta \le \frac{\pi}{18}$:

$$\arg\min_{A_i \in \Delta} \mathrm{dist}_X(A_i, q') = B, \qquad \arg\max_{A_i \in \Delta} \mathrm{dist}_X(A_i, q') = A.$$

Moreover, Lemma 23 also states that $\angle q'AB > \theta$, and hence $q'$ is $\alpha'$-good with respect to the simplex $\Delta$.

The remaining task is to show that $\Delta'_{\alpha'} \subset \Delta_\alpha$.

The largest displacement under the rotation $R_{2\theta}^{-1}$ occurs in the plane $ABQ$. Since all vertices of the simplex, except for $A$, are equidistant from the origin $A$, the point $B$ is therefore the farthest from its image $\gamma(B)$. Hence, to verify that $\Delta'_{\alpha'} \subset \Delta_\alpha$, it suffices to show that

$$B(B', \alpha') \subset B(B, \alpha).$$

Formally, this implication is proved in Lemma 24.

To verify that $B(B', \alpha') \subset B(B, \alpha)$, we apply the Law of Cosines to the triangle $\triangle BAR_{2\theta}(B)$, where both sides $\mathrm{dist}_X(A, B)$ and $\mathrm{dist}_X(A, R_{2\theta}(B))$ equal $2\delta$, and the angle at vertex $A$ is $2\theta$. This gives:

$$\mathrm{dist}_X(B, R_{2\theta}(B))^2 = 4\delta^2 + 4\delta^2 - 8\delta^2 \cos(2\theta) = 8\delta^2(1 - \cos(2\theta)),$$

so

$$\mathrm{dist}_X(B, R_{2\theta}(B)) = 2\delta\sqrt{2(1 - \cos(2\theta))}.$$

Thus, we require:

$$\alpha - \alpha' \ge 2\delta \cdot \sqrt{2 - 2\cos(2\theta)} \quad \Longleftrightarrow \quad a - b \ge \sqrt{2(1 - \cos(2\theta))}.$$

Using the identity $\cos(2\theta) = 2\cos^2\theta - 1 = 2(1 - 2b)^2 - 1$, we compute:

$$1 - \cos(2\theta) = 1 - \left[2(1 - 2b)^2 - 1\right] = 8b(1 - b).$$

Thus, the condition becomes:

$$a - b \ge \sqrt{16b(1 - b)}.$$

Since $a > b$, we can safely square both sides:

$$(a - b)^2 \ge 16b(1 - b) \quad \Longleftrightarrow \quad 17b^2 - (16 + 2a)b + a^2 \ge 0.$$

The discriminant of the quadric polynomial $f(b) = 17b^2 - (16 + 2a)b + a^2$ is

$$\frac{D}{4} = (8 + a)^2 - 17a^2.$$

Since $a < \frac{(8+a)+\sqrt{(8+a)^2-17a^2}}{17}$, the inequality $a - b \geq 4\sqrt{b - b^2}$ holds only when

$$0 < b \leq \frac{(8+a) - \sqrt{(8+a)^2 - 17a^2}}{17}.$$

Finally, note:

$$\frac{(8+a) - \sqrt{(8+a)^2 - 17a^2}}{17} = \frac{a^2}{(8+a) + \sqrt{(8+a)^2 - 17a^2}} > \frac{a^2}{81} = b,$$

which confirms the condition, completing the argument. $\qquad\square$

## F  Technical Results

For completeness, we collect in this appendix several arguments deferred from the main text. We begin in Appendix F.1 with a topological result establishing the right-continuity of $e_X(\alpha)$ on totally bounded metric spaces. In Appendix F.2 we turn to geometric techniques involving rotations of regular simplices, which play an essential role in the proof of Theorem 6. Finally, in Appendix F.3 we give detailed proofs of the illustrative examples that were sketched in Section 2.

### F.1  Right-Continuity of $e_X$

We start by fixing notation. For a subset $S$ in a metric space $(X, \text{dist}_X)$, the *Chebyshev radius* of $S$, denoted $r(S)$, and the *diameter* of $S$, denoted $\text{diam}\, S$, are given by

$$r(S) = \inf_{q \in X} \sup_{x \in S} \text{dist}_X(x, q), \qquad \text{diam}\, S = \sup_{x,y \in S} \text{dist}_X(x, y).$$

Recall that the number $e_X(\alpha)$ is intuitively the maximal radius of an enclosing ball over all sets with diameter at most $\alpha$— and formally defined as

$$e_X(\alpha) := \sup_{\substack{S \subseteq X \\ \text{diam}(S) \leq \alpha}} r(S).$$

Our goal in this section is to show that $e_X$ is right-continuous for every totally bounded metric space $X$.

Let $(X, d)$ be a metric space. Consider the set of compact subsets of the space $X$, denoted by $\mathcal{K}(X)$. Given two nonempty compact subsets $A, B \subseteq X$, their *Hausdorff distance* is defined as

$$\text{dist}_{\mathcal{K}(X)}(A, B) := \max \left\{ \sup_{a \in A} \inf_{b \in B} \text{dist}_X(a, b), \sup_{b \in B} \inf_{a \in A} \text{dist}_X(a, b) \right\}.$$

Intuitively, this measures how far the sets are from being contained in each other's neighborhoods. The space $(\mathcal{K}(X), \text{dist}_{\mathcal{K}(X)})$ is known in the literature as the *hyperspace* of $X$.

A classical result states that $\mathcal{K}(X)$ is compact whenever $X$ is compact (see Theorem 3.5 Illanes and Jr. [1999]). Consequently, every sequence of compact subsets admits a subsequence that converges in the Hausdorff metric, which implies the right-continuity of $e_X(\alpha)$. The proof is given below.

**Lemma 19.** *Let $X$ be a compact metric space. Then the function $e_X(\alpha)$ is right-continuous.*

*Proof.* Observe that any subset $S \subset X$ has the same diameter and Chebyshev radius as its closure. Hence, in the definition of $e_X(\alpha)$, it suffices to consider closed subsets, which are compact since $X$ is compact:

$$e_X((2+\epsilon)\delta) = \sup_{\substack{S \subset X \text{ compact} \\ \text{diam}(S) \leq (2+\epsilon)\delta}} r(S).$$

Both the Chebyshev radius and the diameter are continuous functions on $\mathcal{K}(X)$. Indeed, for any two compact sets $A, B \in \mathcal{K}(X)$,

$$|\text{diam}\, A - \text{diam}\, B| \leq 2 \cdot \text{dist}_{\mathcal{K}(X)}(A, B), \qquad |r(A) - r(B)| \leq \text{dist}_{\mathcal{K}(X)}(A, B).$$

Now take any decreasing sequence $\beta_n \to \beta$. Notice that the function $\mathsf{e}_X(\alpha)$ is non-decreasing, and hence the sequence $\mathsf{e}_X(\beta_n)$ is non-increasing. To prove the result, we need to show that $\mathsf{e}_X(\beta_n) \to \mathsf{e}_X(\beta)$.

Suppose, for contradiction, that there exists $\gamma > 0$ such that for any natural number $n \in \mathbb{N}$,

$$\mathsf{e}_X(\beta_n) > \mathsf{e}_X(\beta) + \gamma.$$

Then, for each natural number $n$, we can find a set $S_n$ of diameter at most $\beta_n$ such that

$$r(S_n) > \mathsf{e}_X(\beta) + \frac{\gamma}{2}.$$

Since $\mathcal{K}(X)$ is compact, there exists a convergent subsequence of compacts subsets $S_{m(n)} \to S$ in the Hausdorff metric. Continuity of the diameter implies $\operatorname{diam} S \le \beta$, and continuity of the Chebyshev radius gives $r(S_{m(n)}) \to r(S)$. Hence,

$$r(S) \ge \mathsf{e}_X(\beta) + \frac{\gamma}{2},$$

contradicting the definition of $\mathsf{e}_X(\beta)$. This proves the claim. $\qquad\square$

**Lemma 20.** *For a totally bounded metric space $X$, the function $\mathsf{e}_X$ is right-continuous.*

*Proof.* A standard extension of the Heine–Borel theorem states that a metric space is compact if and only if it is complete and totally bounded (see Munkres [2000][Theorem 45.1]). The completion $\hat{X}$ of a totally bounded space $X$ is complete by construction and remains totally bounded, hence compact. Because $X$ is dense in $\hat{X}$, every subset of $X$ can be approximated arbitrarily well by subsets of $\hat{X}$ (and vice-versa), so $\mathsf{e}_X = \mathsf{e}_{\hat{X}}$. Lemma 19 now applies to $\hat{X}$, yielding the desired right-continuity for $\mathsf{e}_X$. $\qquad\square$

## F.2   Geometric tools for Euclidean simplices

This section presents the geometric constructions that, while essential, would otherwise disrupt the logical flow of Theorem 6 in which they are applied.

Let $\Delta = \{A_i\}_{i=0}^n$ be a regular simplex[10] in $\mathbb{R}^n$ with edge length $2\delta$. To distinguish two specific vertices, set

$$A := A_1, \qquad B := A_2.$$

Let $Q$ denote the centroid of the remaining vertices:

$$Q = \frac{1}{n-1} \sum_{C \in \Delta \setminus \{A,B\}} C.$$

Consider the rotation

$$R_{2\theta} : \mathbb{R}^n \to \mathbb{R}^n$$

that fixes the vertex $A$, acts in the plane $\Pi = \operatorname{span}\{A, B, Q\}$ as a rotation by angle $2\theta$ around $A$ in the direction from $\overrightarrow{AB}$ to $\overrightarrow{AQ}$ along the smaller angle between them, and acts as the identity on the orthogonal complement $\Pi^\perp$.

If we place the point $A$ at the origin and choose an orthonormal basis $\vec{d}_1, \ldots, \vec{d}_n$ such that $\operatorname{span}\{\vec{d}_1, \vec{d}_2\} = \Pi$, and

$$\overrightarrow{AB} = k \cdot \vec{d}_1 \quad \text{with } k > 0, \qquad \overrightarrow{AQ} = k_1 \vec{d}_1 + k_2 \vec{d}_2 \quad \text{with } k_1, k_2 > 0,$$

then[11], the rotation $R_{2\theta}$ is represented in this basis by the matrix

$$R_{2\theta} = \begin{pmatrix} \cos(2\theta) & -\sin(2\theta) & 0 & \cdots & 0 \\ \sin(2\theta) & \cos(2\theta) & 0 & \cdots & 0 \\ 0 & 0 & 1 & & 0 \\ \vdots & \vdots & & \ddots & \\ 0 & 0 & 0 & & 1 \end{pmatrix}. \tag{4}$$

---

[10]We use $\Delta$ to denote the discrete set of $n+1$ vertices of the regular simplex, rather than its convex hull.
[11]Notice that such a basis exists because $\angle BAQ \le \frac{\pi}{2}$.

The upper-left $2 \times 2$ block corresponds to a counterclockwise rotation in the plane $\Pi$, and the rest acts as the identity on $\Pi^\perp$.

This transformation is a Euclidean isometry: it preserves all distances and acts as a rotation in the $ABQ$-plane while leaving the orthogonal directions unchanged.

**Remark 21.** *Given two vectors $v_1 := \overrightarrow{NM}$ and $v_2 := \overrightarrow{NK}$ for some points $\{N, M, K\}$, we will often refer to $\cos \angle MNK$ as $\cos(v_1, v_2)$. This emphasizes the computational role of the cosine as the inner product between the normalized vectors $v_1$ and $v_2$:*

$$\cos \angle MNK = \frac{v_1}{\|v_1\|} \cdot \frac{v_2}{\|v_2\|}.$$

The following lemma is a basic yet useful observation.

**Lemma 22.** *Let $\Pi$ be a plane, and let $A, B \in \Pi$ be two distinct points. Fix any point $q \in \mathbb{R}^n$, distinct from $A$, and let $H_q$ denote the orthogonal projection of $q$ onto $\Pi$. Then,*

$$\begin{cases} 0 < \cos \angle qAB \leq \cos \angle H_q AB, & \text{if } \angle qAB < \frac{\pi}{2}, \\ 0 > \cos \angle qAB \geq \cos \angle H_q AB, & \text{if } \angle qAB > \frac{\pi}{2}, \\ \cos \angle qAB = \cos \angle H_q AB = 0, & \text{if } \angle qAB = \frac{\pi}{2}. \end{cases}$$

*Proof.* Since the cosine between two vectors can be computed via their inner product (see Remark 21), the key observation is that

$$\overrightarrow{AB} \cdot \overrightarrow{Aq} = \overrightarrow{AB} \cdot \overrightarrow{AH_q},$$

because the vectors $\overrightarrow{AB}$ and $\overrightarrow{H_q q}$ are orthogonal. Indeed, since $H_q$ is the projection of $q$ onto the plane $\Pi$, the vector $\overrightarrow{H_q q}$ is orthogonal to every vector in $\Pi$, including $\overrightarrow{AB}$.

Hence,

$$\overrightarrow{AB} \cdot \overrightarrow{Aq} = \overrightarrow{AB} \cdot \overrightarrow{AH_q}.$$

Now, if $H_q \neq A$ (the case $H_q = A$ is trivial), we compute:

$$\begin{aligned} \cos(\overrightarrow{Aq}, \overrightarrow{AB}) &= \frac{\overrightarrow{Aq}}{\|Aq\|} \cdot \frac{\overrightarrow{AB}}{\|AB\|} = \frac{\overrightarrow{AH_q} \cdot \overrightarrow{AB}}{\|Aq\| \cdot \|AB\|} \\ &= \left( \frac{\|AH_q\|}{\|Aq\|} \right) \cdot \left( \frac{\overrightarrow{AH_q}}{\|AH_q\|} \cdot \frac{\overrightarrow{AB}}{\|AB\|} \right) \\ &= \cos(\overrightarrow{AH_q}, \overrightarrow{AB}) \cdot \frac{\|AH_q\|}{\|Aq\|}. \end{aligned}$$

Since $AH_q$ is the projection of $Aq$ onto the plane $\Pi$, we have

$$0 \leq \frac{\|AH_q\|}{\|Aq\|} \leq 1,$$

with equality only if $q \in \Pi$.

Now observe that for any angle $\phi \leq \pi$,

$$\cos \phi < 0 \quad \Longleftrightarrow \quad \phi > \frac{\pi}{2}.$$

Thus, the product $\cos(\overrightarrow{AH_q}, \overrightarrow{AB}) \cdot \frac{\|AH_q\|}{\|Aq\|}$ is: - smaller than $\cos(\angle H_q AB)$ if $\angle qAB < \frac{\pi}{2}$, - larger if $\angle qAB > \frac{\pi}{2}$, and - equal when the angle is $\frac{\pi}{2}$, since then both cosines vanish.

This completes the proof. $\square$

**Lemma 23** (Rotation in the $ABQ$-plane keeps near/far order)**.** *Assume for a query point $q \in \mathbb{R}^n$, the nearest and farthest points are $B$ and $A$ respectively; that is:*

$$B = \operatorname{argmin}_{A_i \in \Delta} \operatorname{dist}_X(q, A_i), \qquad A = \operatorname{argmax}_{A_i \in \Delta} \operatorname{dist}_X(q, A_i).$$

*Then, for any angle $\theta < \frac{\pi}{18}$ whenever $\angle BAq \leq \theta$ the isometry $R_{2\theta}$ preserves both the nearest and the farthest vertices of $\Delta$ with respect to $q$:*

$$B = \operatorname{argmin}_{A_i \in \Delta} \operatorname{dist}_X(R_{2\theta}q, A_i), \qquad A = \operatorname{argmax}_{A_i \in \Delta} \operatorname{dist}_X(R_{2\theta}q, A_i).$$

*Moreover, the rotated point $q' := R_{2\theta}q$ satisfies*

$$\angle q'AB > \theta.$$

*Proof.* We will show that the nearest point $B$ remains the nearest, and that the farthest point $A$ likewise remains the farthest, whenever $\angle qAB \leq \frac{\pi}{18}$. Finally, we will establish that $\angle q'AB > \theta$.

Our argument will be carried out in an explicit orthonormal basis adapted to the geometry of the simplex.

**Coordinates in an orthonormal basis.**

To simplify the calculations, we first scale the simplex $A_1 A_2 \ldots A_{n+1}$ by a factor of $1/(2\delta)$. We then embed it in $\mathbb{R}^{n+1}$ by mapping the $i^{\text{th}}$ vertex to $\frac{1}{\sqrt{2}}e_i$. Throughout, we set $A := A_1$ and $B := A_2$. In these coordinates,

$$\overrightarrow{AB} = \tfrac{1}{\sqrt{2}}(-1, 1, 0, \ldots, 0), \qquad \overrightarrow{AQ} = \tfrac{1}{\sqrt{2}}\left(-1, 0, \tfrac{1}{n-1}, \ldots, \tfrac{1}{n-1}\right).$$

Next, we introduce the two unit vectors spanning the affine plane $ABQ$, which will play a key role in defining the rotation $R_{2\theta}$ (see (4)).

Writing the plane as

$$ABQ = A + \langle \vec{d_1}, \vec{d_2} \rangle,$$

with Minkowski addition and linear span, we take

$$\vec{d_1} := \overrightarrow{AB}, \qquad \vec{d_2} := \sqrt{\tfrac{n-1}{2(n+1)}}\left(-1, -1, \tfrac{2}{n-1}, \ldots, \tfrac{2}{n-1}\right).$$

A direct check confirms that $\{\vec{d_1}, \vec{d_2}\}$ is orthonormal. Moreover

$$2\overrightarrow{AQ} - \overrightarrow{AB} = \sqrt{\tfrac{n+1}{n-1}}\,\vec{d_2}, \qquad \implies \qquad 2\overrightarrow{AQ} = \sqrt{\tfrac{n+1}{n-1}}\,\vec{d_2} + \vec{d_1}.$$

*Completing the basis.* Pick an orthonormal completion $\{\vec{d_i}\}_{i=3}^{n}$ of $\{\vec{d_1}, \vec{d_2}\}$. A convenient choice is

$$\vec{d_3} := \frac{\overrightarrow{QA_3}}{\|QA_3\|} = \gamma\left(0, 0, \tfrac{n-2}{n-1}, -\tfrac{1}{n-1}, \ldots, -\tfrac{1}{n-1}\right), \quad \gamma := \sqrt{\tfrac{n-1}{n-2}},$$

and analogous definitions for $i \geq 4$.

With this basis, for any $3 \leq i \leq n+1$ we have

$$\overrightarrow{AA_i} = \tfrac{1}{2}\left(\sqrt{\tfrac{2(n-2)}{n-1}}\,\vec{d_i} + \sqrt{\tfrac{n+1}{n-1}}\,\vec{d_2} + \vec{d_1}\right).$$

**Coordinates of the query point.** Let $q$ satisfy $\angle qAB \leq \theta$ and write

$$\overrightarrow{Aq} = \sum_{i=1}^{n} z_i \vec{d_i}, \quad \text{so that} \quad \frac{z_1}{\sqrt{\sum_{i=1}^n z_i^2}} = \cos(\angle qAB) \geq \cos\theta. \tag{5}$$

In particular, $z_1 \geq (\cos\theta/\sin\theta)\,|z_2|$.

Observe that for any vertex $A_i$ of the simplex and for *any* point $C$, one has

$$|AC| \geq |A_iC| \iff 2\overrightarrow{AC} \cdot \overrightarrow{AA_i} \geq 1, \tag{6}$$

since

$$\|\overrightarrow{A_iC}\|^2 = \|\overrightarrow{AC} - \overrightarrow{AA_i}\|^2 = \|\overrightarrow{AC}\|^2 - 2\overrightarrow{AC} \cdot \overrightarrow{AA_i} + 1.$$

Moreover, for any point $C$ with coordinates $\overrightarrow{AC} = \sum_{i=1}^{n} y_i d_i$, a straightforward calculation shows that

$$\overrightarrow{AC} \cdot \overrightarrow{AA_i} = \tfrac{1}{2}\left(\sqrt{\tfrac{2(n-2)}{n-1}}\, y_i + \sqrt{\tfrac{n+1}{n-1}}\, y_2 + y_1\right). \tag{7}$$

**Step I: the farthest point remains the farthest when $\angle qAB \leq \frac{\pi}{8}$.**

Because $R_{2\theta}$ acts as a planar rotation in $\langle \vec{d_1}, \vec{d_2}\rangle$ and fixes the orthogonal complement, we have

$$\overrightarrow{Aq'} = (\cos 2\theta\, z_1 - \sin 2\theta\, z_2)\vec{d_1} + (\cos 2\theta\, z_2 + \sin 2\theta\, z_1)\vec{d_2} + \sum_{i=3}^{n} z_i\, \vec{d_i}.$$

Fix any $n + 1 \geq i \geq 3$. By Eq. (6) and Eq. (7), to show that $|Aq| \geq |A_i q|$ implies $|Aq'| \geq |A_i q'|$, it suffices to verify that the inequality

$$\sqrt{\tfrac{2(n-2)}{n-1}}\, z_i + \sqrt{\tfrac{n+1}{n-1}}\, z_2 + z_1 \;\geq\; 1$$

implies

$$\sqrt{\tfrac{2(n-2)}{n-1}}\, z_i + \sqrt{\tfrac{n+1}{n-1}}\big(\cos 2\theta\, z_2 + \sin 2\theta\, z_1\big) + \big(\cos 2\theta\, z_1 - \sin 2\theta\, z_2\big) \;\geq\; 1.$$

For this implication to hold, it suffices to verify

$$\left(\sqrt{\tfrac{n+1}{n-1}} \sin 2\theta + \cos 2\theta - 1\right) z_1 \;\geq\; \left(\sqrt{\tfrac{n+1}{n-1}}(1 - \cos 2\theta) + \sin 2\theta\right) z_2,$$

or, equivalently (using the double-angle identities),

$$\left(\sqrt{\tfrac{n+1}{n-1}} \cos\theta - \sin\theta\right) z_1 \;\geq\; \left(\sqrt{\tfrac{n+1}{n-1}} \sin\theta + \cos\theta\right) z_2.$$

Because $z_1 \geq (\cos\theta/\sin\theta)|z_2|$ (see Eq. (5)), it is enough to verify

$$\left(\sqrt{\tfrac{n+1}{n-1}} \cos\theta - \sin\theta\right) \tfrac{\cos\theta}{\sin\theta} \;\geq\; \sqrt{\tfrac{n+1}{n-1}} \sin\theta + \cos\theta,$$

which in turn is equivalent to

$$\sqrt{\tfrac{n+1}{n-1}} \cos 2\theta - \sin 2\theta \;\geq\; 0.$$

The latter holds for every $0 < \theta \leq \frac{\pi}{8}$, completing the proof.

**Step II: the nearest point remains the nearest when $\angle qAB \leq \frac{\pi}{18}$.**

We will prove that whenever $\angle q'AB \leq \frac{\pi}{6}$, all distances $|A_i q'|$ for $A_i \notin \{A, B\}$ are greater than $|Bq'|$. To that end, let us denote the angle $\angle q'AB$ by $\phi$ and fix any $n + 1 \geq i \geq 3$.

By Eq. (6), the claim is equivalent to the scalar-product inequality

$$\overrightarrow{Aq'} \cdot \overrightarrow{BA} \;\leq\; \overrightarrow{Aq'} \cdot \overrightarrow{A_i A},$$

and by Eq. (7), in our orthonormal coordinates, this becomes

$$\overrightarrow{Aq'} \cdot \overrightarrow{AB} = z_1 \;\geq\; \tfrac{1}{2}\left(\sqrt{\tfrac{2(n-2)}{n-1}}\, z_i + \sqrt{\tfrac{n+1}{n-1}}\, z_2 + z_1\right) = \overrightarrow{Aq'} \cdot \overrightarrow{AA_i},$$

or, equivalently,

$$z_1 \;\geq\; \sqrt{\tfrac{2(n-2)}{n-1}}\, z_i + \sqrt{\tfrac{n+1}{n-1}}\, z_2. \tag{8}$$

Assume $\angle q'AB = \phi \leq \pi/6$. Then, by Eq. (5),

$$z_1 \;\geq\; \frac{\cos\phi}{\sin\phi} \sqrt{z_2^2 + z_i^2} \;\geq\; \sqrt{3} \cdot \sqrt{z_2^2 + z_i^2},$$

where the last inequality uses $\frac{\cos\phi}{\sin\phi} \geq \frac{\cos(\pi/6)}{\sin(\pi/6)} = \sqrt{3}$, which holds for all $0 < \phi \leq \pi/6$.

We will now prove that

$$3z_2^2 + 3z_i^2 \geq \left( \sqrt{\tfrac{2(n-2)}{n-1}}\, z_i + \sqrt{\tfrac{n+1}{n-1}}\, z_2 \right)^2.$$

Expanding the right-hand side, this is equivalent to

$$\frac{n+1}{n-1}z_i^2 + \frac{2(n-2)}{n-1}z_2^2 \geq 2\sqrt{\frac{2(n-2)(n+1)}{(n-1)^2}}\, z_i z_2,$$

which is always true by the inequality $U^2 + S^2 \geq 2US$, applied with

$$U = \sqrt{\frac{n+1}{n-1}}\, z_i, \qquad S = \sqrt{\frac{2(n-2)}{n-1}}\, z_2.$$

Putting these estimates together, we obtain

$$z_1 \geq \sqrt{3z_2^2 + 3z_i^2} \geq \sqrt{\tfrac{2(n-2)}{n-1}}\, z_i + \sqrt{\tfrac{n+1}{n-1}}\, z_2,$$

which is exactly (8). Hence $|Bq'| \leq |A_i q'|$ whenever $\angle q'AB \leq \pi/6$.

*Consequence for the rotated point.* If the rotation parameter satisfies $\theta \leq \pi/18$, then the rotated point $q'$ obeys $\angle BAq' \leq \pi/6$; therefore the point $B$ remains the nearest after rotation.

Hence, if the angle $\theta \leq \frac{\pi}{18}$, the angle $\angle BAq' \leq \frac{\pi}{6}$ and the point $B$ still remains the nearest.

**Step III: proving $\angle q'AB > \theta$**

It remains to verify that the rotation $R_{2\theta}$ sufficiently increases the angle, i.e., that $\angle BAq' \geq \theta$.

To show that decompose the vector $\overrightarrow{Aq}$ as

$$\overrightarrow{Aq} = \overrightarrow{AH_q} + \overrightarrow{H_q q},$$

where $H_q$ is the projection of $q$ onto the plane $\Pi$. Denote $R_{2\theta}[H_q]$ by $H_q'$. Since $R_{2\theta}$ is not only an isometry, but by construction also a linear transformation with placing the point $A$ as the origin:

$$\overrightarrow{Aq'} = R_{2\theta}[\overrightarrow{Aq}] = R_{2\theta}[\overrightarrow{AH_q}] + R_{2\theta}[\overrightarrow{H_q q}] =$$
$$\overrightarrow{AH_q'} + R_{2\theta}[\overrightarrow{H_q q}].$$

Since the rotation $R_{2\theta}$ preserves the vectors orthogonal to $\Pi$, it also preserves $\overrightarrow{H_q q}$. Hence (see Fig. 7):

$$\overrightarrow{Aq'} = \overrightarrow{AH_q'} + \overrightarrow{H_q q},$$

and moreover, $H_q'$ is the projection of $q'$ onto the plane $\Pi$.

To see that $\angle q'AB \geq \theta$, note that the cosine function is decreasing on the interval $[0, \pi]$. Therefore, it suffices to show that

$$\cos(\angle q'AB) \leq \cos\theta.$$

For this purpose, we will use Lemma 22, which formalizes the observation that projecting a point onto a plane either increases or decreases the cosine of an angle, depending on whether the angle is acute or obtuse. Specifically, if $\angle qAB < \frac{\pi}{2}$, then $\cos\angle qAB \geq \cos\angle H_q AB$, and the inequality is reversed when the angle is obtuse.[12] However, to apply this lemma correctly, one must verify that either the original angle or its projection is acute or obtuse, as the conclusion depends on this distinction.

By Lemma 22, we have $\cos\angle H_q AB \geq \cos\angle qAB$, since $\angle qAB \leq \theta < \frac{\pi}{2}$. Therefore, using again that cosine is decreasing on the interval $[0, \pi]$, it follows that $\angle H_q AB \leq \theta$.

---

[12]It also shows that the projected angle is acute or obtuse if and only if the original angle was acute or obtuse.

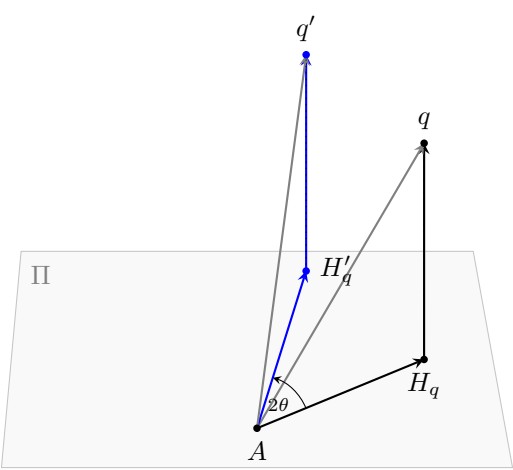

Figure 7: Rotation and vector decomposition of $\overrightarrow{Aq}$.

Using the triangle inequality for angles, and the fact that $\angle H_q A H'_q = 2\theta$, we deduce:

$$\angle H'_q AB \le \angle H_q A H'_q + \angle H_q AB \le 2\theta + \theta = 3\theta < \frac{\pi}{2},$$

$$\angle H'_q AB \ge \angle H_q A H'_q - \angle H_q AB \ge 2\theta - \theta = \theta.$$

Therefore, applying Lemma 22 once again, we conclude:

$$\cos \angle q' AB \le \cos \angle H'_q AB \le \cos \theta.$$

$\square$

**Lemma 24.** *Suppose $\alpha', \alpha > 0$ are such that $R_{2\theta}^{-1}\big[B(B, \alpha')\big] \subset B(B, \alpha)$. Then*

$$R_{2\theta}^{-1}\big[\Delta_{\alpha'}\big] \subset \Delta_\alpha.$$

*Proof.* Denote the simplex $R_{2\theta}^{-1}[\Delta]$ by $\Delta'$. To ensure the inclusion $\Delta'_{\alpha'} \subset \Delta_\alpha$, it suffices to check that no vertex of the simplex moves by more than $\alpha - \alpha'$ under the rotation $R_{2\theta}$.

Let $C$ be any vertex distinct from $A$ and $B$. Then the point $C$ does not lie in the affine plane $ABQ$. Denote by $B' := R_{2\theta}B$, by $C' := R_{2\theta}C$, and by $Q' := R_{2\theta}Q$. The point $C$ is projected onto $Q$.[13] Then,

$$\cos(\overrightarrow{AQ}, \overrightarrow{AQ'}) = \frac{\overrightarrow{AQ} \cdot \overrightarrow{AQ'}}{\|AQ\|^2} = \cos 2\theta,$$

due to the construction of the rotation. On the other hand, since $\overrightarrow{AQ} \perp \overrightarrow{QC}$, one has:

$$\cos(\overrightarrow{AC}, \overrightarrow{AC'}) = \frac{\overrightarrow{AQ} \cdot \overrightarrow{AQ'} + \|CQ\|^2}{\|AQ\|^2 + \|CQ\|^2} \ge \frac{\overrightarrow{AQ} \cdot \overrightarrow{AQ'}}{\|AQ\|^2} = \cos 2\theta,$$

since adding the same positive value to both the numerator and denominator of a ratio in $(0, 1]$ does not decrease the ratio. Now,

$$\begin{aligned}
\|BB'\|^2 - \|CC'\|^2 &= \|\overrightarrow{AB} - \overrightarrow{AB'}\|^2 - \|\overrightarrow{AC} - \overrightarrow{AC'}\|^2 \\
&= 2\left(\overrightarrow{AC} \cdot \overrightarrow{AC'} - \overrightarrow{AB} \cdot \overrightarrow{AB'}\right) \\
&= 2\|\overrightarrow{AB}\|^2\left(\cos(\overrightarrow{AC}, \overrightarrow{AC'}) - \cos 2\theta\right) > 0,
\end{aligned}$$

which shows that indeed $\|CC'\| < \|BB'\|$. $\square$

---

[13]But this fact is not essential; the argument still holds without requiring that the projection coincides with $Q$.

### F.3 Examples

We begin by analyzing the game on the real line.

**Example** (Example 7 from the Introduction). *Let $\epsilon > 0$ and $\delta \geq 0$. Then, for any number $T$ of queries,*

$$\mathrm{OPT}_{\mathbb{R}}(T, \epsilon, \delta) = +\infty.$$

*Proof.* The idea of the proof is straightforward: since the space is unbounded, the responder can—already in the first round—return an arbitrarily large answer. This ensures that the initial feasible region is as large as desired. Then, over the course of $T$ interactions, the responder can control how fast the region shrinks, ensuring that the final feasible region remains arbitrarily large.

Let us elaborate.

At the start of the game, for any large number $L_0 > 0$ and any query $q \in \mathbb{R}$, the responder may answer with

$$r_q := \frac{1 + \epsilon}{(1 + \epsilon)^2 - 1} \left(L_0 - (2 + \epsilon)\delta\right),$$

which results in a feasible region that includes two intervals of length $L_0$.

Now fix an interval $[a, b]$ of length $L$, and suppose the reconstructor asks a query $q \in \mathbb{R}$. The responder then answers with

$$r_q := \frac{\max\{|q - b|, |q - a|\} - \delta}{1 + \epsilon}.$$

This response places the point in $[a, b]$ that is farthest from $q$ right on the boundary of the feasible region $\Phi(q, r_q)$. In particular, this implies that every point $x \in [a, b]$ satisfies $|x - q| \leq (1 + \epsilon)r_q + \delta$.

Assume without loss of generality that $\max\{|q - b|, |q - a|\} = |q - b|$, i.e., $q \leq \frac{a+b}{2}$. On the other hand, all points $x \in [a, b]$ satisfying

$$|x - q| \geq \frac{|b - q| - (2 + \epsilon)\delta}{(1 + \epsilon)^2}$$

also satisfy $(1 + \epsilon) \cdot |x - q| + \delta \geq r_q$. Thus, all such points lie within $\Phi(q, r_q)$.

The length of the subinterval of $[a, b]$ consisting of such points is

$$\frac{((1 + \epsilon)^2 - 1)|b - q| + (2 + \epsilon)\delta}{(1 + \epsilon)^2} \geq \frac{((1 + \epsilon)^2 - 1) \cdot |b - a|}{2(1 + \epsilon)^2} = \frac{((1 + \epsilon)^2 - 1)}{2(1 + \epsilon)^2} \cdot |b - a|.$$

Hence, on each round, the responder can reduce the feasible region's length by a constant multiplicative factor $c := \frac{(1+\epsilon)^2 - 1}{2(1+\epsilon)^2}$. Starting from an interval of arbitrary length $L_0$, the feasible region after $T$ rounds can still have length at least $c^T \cdot L_0$, which diverges as $L_0 \to \infty$.

Therefore,

$$\mathrm{OPT}_{\mathbb{R}}(T, \epsilon, \delta) = +\infty.$$

$\square$

The next example demonstrates that when $\epsilon = 0$, the real line is $(\epsilon, \delta)-$pseudo-finite for every $\delta > 0$:

**Example 25** (Pseudo-finiteness on the real line). *For every $\delta \geq 0$, the real line $\mathbb{R}$ with its usual metric is $(0, \delta)$-pseudo-finite.*

*Proof.* The optimal reconstructor strategy is to ask two query points $q_1, q_2 \in \mathbb{R}$ with $q_2 - q_1 > 2\delta$. Let the answers of the responder be $r_1, r_2$.

Intuitively, each answer restricts the secret to intervals of length $2\delta$ centered at $q_i \pm r_i$. Because the distance between the center of the leftmost interval and the center of the rightmost interval exceeds $2\delta$, at most two of the four candidate intervals overlap, and their intersection has diameter $2\delta$, attaining the optimal error (see Figure 8).

Formally, the feasible regions are

$$\Phi(q_1, r_1) = B\left(q_1 - r_1, \delta\right) \cup B\left(q_1 + r_1, \delta\right),$$
$$\Phi(q_2, r_2) = B\left(q_2 - r_2, \delta\right) \cup B\left(q_2 + r_2, \delta\right).$$

Assume there are two points $x, y$ in the intersection $\Phi(q_1, r_1) \cap \Phi(q_2, r_2)$, such that $y - x > 2\delta$. These points cannot lie in the same ball of radius $\delta$, hence

$$x \in B\left(q_1 - r_1, \delta\right), \quad y \in B\left(q_1 + r_1, \delta\right),$$

and also

$$x \in B\left(q_2 - r_2, \delta\right), \quad y \in B\left(q_2 + r_2, \delta\right).$$

Therefore, the balls with larger and smaller centers must overlap:

$$|q_2 + r_2 - (q_1 + r_1)| < 2\delta, \quad \text{and} \quad |q_2 - r_2 - (q_1 - r_1)| < 2\delta.$$

On the other hand,

$$|q_2 - q_1 + r_2 - r_1| + |q_2 - q_1 + r_1 - r_2| \geq 2|q_2 - q_1| > 4\delta,$$

which leads to a contradiction. Hence the result. $\qquad\square$

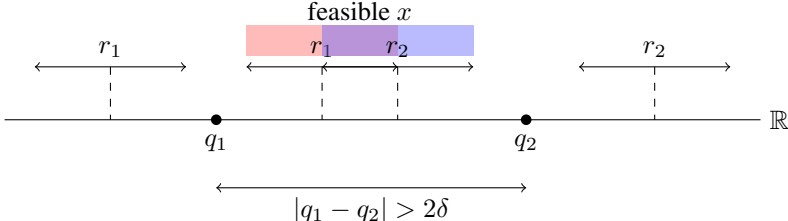

Figure 8: With $|q_1 - q_2| > 2\delta$, only the red interval $[q_1 + r_1 - \delta,\ q_1 + r_1 + \delta]$ and the blue interval $[q_2 - r_2 - \delta,\ q_2 - r_2 + \delta]$ intersect, pinning the secret point to their (purple) overlap.

The following simple observation shows that in bounded metric spaces, the reconstruction game becomes trivial whenever $(2 + \epsilon)\delta$ exceeds the diameter of the space.

**Example 26.** *Any bounded metric space $X$ with $\mathrm{diam}(X) \leq (2 + \epsilon)\delta$ is $(\epsilon, \delta)$-pseudo-finite. Indeed, in this regime, the responder can maintain the entire space as feasible throughout the interaction by consistently replying with the constant value $\delta$. As a result, the optimal reconstruction error is simply the Chebyshev radius of $X$, which the reconstructor can achieve without submitting any queries.*

The next two examples concern noiseless responders (i.e., $\epsilon = \delta = 0$):

**Example 27.** *The Euclidean space $\mathbb{R}^n$ is $(0,0)$-pseudo-finite. Indeed, any point $x \in \mathbb{R}^n$ is uniquely determined by its distances to the $n + 1$ vertices of a non-degenerate $n$-simplex [Blumenthal, 1970, §2].*

The same holds for any subset of $\mathbb{R}^n$ that contains such a simplex. However, even in the noiseless setting $(\epsilon, \delta) = (0, 0)$, pseudo-finiteness does not hold in all metric spaces—even if the space is totally bounded:

**Example 28.** *Let $X = \{0, 1\}^{\mathbb{N}}$ be the space of infinite binary sequences, equipped with the standard ultrametric: the distance between two sequences $\alpha = (\alpha_i)_{i \in \mathbb{N}}$ and $\beta = (\beta_i)_{i \in \mathbb{N}}$ is defined as $d(\alpha, \beta) = 2^{-j}$, where $j$ is the first index for which $\alpha_j \neq \beta_j$. Then $X$ is a compact (and hence totally bounded) metric space that is not $(0, 0)$-pseudo-finite.*

*Proof.* We show that $\mathrm{OPT}_X(T, 0, 0) \geq 2^{-T-1}$ for every $T$, by explicitly constructing a responder strategy. The goal is to preserve a feasible set of sequences that agree on at most $T$ coordinates.

Assume that after round $t$, the responder has committed to at most the first $t' \leq t$ bits of the secret sequence. Given a query $q = (q_i)_{i \in \mathbb{N}}$, the responder replies as follows:

- If the prefix $(q_1, \ldots, q_{t'})$ disagrees with the committed prefix, respond with the true distance $2^{-j}$, where $j$ is the first index of disagreement.

- Otherwise, respond with $r = 2^{-t'-1}$, and define the next bit of the secret sequence as $\alpha_{t'+1} := 1 - q_{t'+1}$.

After $T$ rounds, the responder has specified exactly $T$ bits. Let the reconstructor return a sequence $\hat{x}$. Then the responder chooses a secret point $x^\star$ that agrees with $\hat{x}$ on all bits except for bit $T+1$, which is flipped. This implies that $\operatorname{dist}_X(\hat{x}, x^\star) = 2^{-T-1}$, yielding the lower bound.

*Remark:* One can further show that this lower bound is tight, and that $\operatorname{OPT}_X(T, 0, 0) = 2^{-T-1}$, since every informative query forces the responder to reveal one additional bit. □

We now present an example of a non–totally bounded metric space for which the function $\mathsf{e}_X$ fails to be right-continuous.

**Example 29** (Failure of right-continuity of $\mathsf{e}_X$). *Recall that for a metric space $(X, \operatorname{dist}_X)$ the function $\mathsf{e}_X$ is defined by*

$$\mathsf{e}_X(\alpha) := \sup\{\, r(S) \; : \; S \subseteq X, \; \operatorname{diam}(S) \leq \alpha \,\},$$

*where the Chebyshev radius and diameter are*

$$r(S) := \inf_{q \in X} \sup_{x \in S} \operatorname{dist}_X(x, q), \qquad \operatorname{diam}(S) := \sup_{x, y \in S} \operatorname{dist}_X(x, y).$$

*Let $X = \{x_n, y_n : n \in \mathbb{N}\}$ with metric*

$$\operatorname{dist}_X(x_n, y_n) = 1 + \tfrac{1}{n} \quad \text{for each } n, \qquad \operatorname{dist}_X(u, v) = 2 \text{ for all other distinct } u \neq v.$$

*Then $\mathsf{e}_X$ is not right-continuous[14].*

*Proof.* If $\alpha \leq 1$, then any subset $S \subseteq X$ with $\operatorname{diam}(S) \leq \alpha$ must be a singleton (since every nontrivial distance is $> 1$), hence $\mathsf{e}_X(\alpha) = 0$.

For each $n$, let $S_n = \{x_n, y_n\}$. Then $\operatorname{diam}(S_n) = 1 + \tfrac{1}{n}$. Moreover,

$$r(S_n) = \inf_{q \in X} \max\{\operatorname{dist}_X(x_n, q), \operatorname{dist}_X(y_n, q)\} = \min\{1 + \tfrac{1}{n}, \, 2\} = 1 + \tfrac{1}{n},$$

because choosing $q \in \{x_n, y_n\}$ yields value $1 + \tfrac{1}{n}$, while any $q \notin S_n$ is at distance 2 from both points.

Note that any subset of $X$ with at least three distinct points contains two points at distance 2, hence has diameter 2. Therefore, for $1 + \tfrac{1}{n} \leq \alpha < 1 + \tfrac{1}{n-1}$ the only nontrivial subsets with $\operatorname{diam} \leq \alpha$ are the pairs $S_k$ with $k \geq n$, and thus

$$\mathsf{e}_X(\alpha) = \max_{k \geq n} r(S_k) = 1 + \tfrac{1}{n}.$$

Consequently,

$$\lim_{\alpha \downarrow 1} \mathsf{e}_X(\alpha) = \lim_{n \to \infty} \mathsf{e}_X\left(1 + \tfrac{1}{n}\right) = 1, \qquad \text{while} \qquad \mathsf{e}_X(1) = 0,$$

so $\mathsf{e}_X$ is not right-continuous at $\alpha = 1$. □

We conclude the section by formally proving the equivalence between the Dinur–Nissim model and the reconstruction game on the Boolean cube as referenced in Example 1.

**Example 30** (Dinur–Nissim model). *The counting-query game in the Dinur–Nissim model is equivalent to the distance-based game on the Boolean cube with the Hamming metric, namely, every query in one game can be simulated by at most two queries in the other.*

---

[14]The space $X$ is not totally bounded: for $\alpha \leq 1$ every $\alpha$-ball contains at most one point (all nonzero distances in $X$ exceed 1), so no finite $\alpha$-net exists; equivalently, the only $\alpha$-cover is $X$ itself, which is infinite.

*Proof.* We show that the counting-query game is equivalent to the distance-based game on the Boolean cube (with Hamming distance) by introducing an intermediate step: both games are equivalent to an inner-product game played on $\{\pm 1\}^n$.

The inner-product game is defined as follows. The responder chooses a secret vector $D' = (d'_1, \ldots, d'_n) \in \{\pm 1\}^n$. In each round, the reconstructor submits a query vector

$$w = (w_1, \ldots, w_n) \in \{\pm 1\}^n,$$

and the responder replies with a noisy approximation of the inner product

$$\langle D', w \rangle = \sum_{i=1}^{n} w_i d'_i.$$

**Step I: From the Dinur–Nissim model to the inner-product game.** In the Dinur–Nissim model, the dataset is a binary vector $D = (d_1, \ldots, d_n) \in \{0, 1\}^n$, and each query is a subset $q \subseteq [n]$, whose (noisy) answer is the count

$$a_q = \sum_{i \in q} d_i.$$

We can represent the subset $q$ by its indicator vector $v_q \in \{0, 1\}^n$, so that $a_q = \langle D, v_q \rangle$. To simulate this count using the inner-product game on $\{\pm 1\}^n$, consider the transformation

$$v \mapsto 2v - 1,$$

which maps $\{0, 1\}^n$ to $\{\pm 1\}^n$. Let

$$D' = 2D - 1 \quad \text{and} \quad w_q = 2v_q - 1.$$

Then we have the identity

$$\langle D', w_q \rangle = 4\langle D, v_q \rangle - 2\langle D, \mathbf{1} \rangle - 2|q| + n.$$

Therefore, we can recover the original count $\langle D, v_q \rangle$ by submitting two inner-product queries: one with $w_q$ and one with the all-ones vector $\mathbf{1}$. A similar argument gives the reverse direction.

**Step II: From the inner-product game to the distance-based game.** Next, we show that the inner-product game on $\{\pm 1\}^n$ is equivalent to the distance game on $\{\pm 1\}^n$ equipped with the Hamming metric. On this space, one has the identity

$$\mathrm{dist}_{\mathrm{Ham}}(x, y) = \frac{1}{4}\|x - y\|_2^2 = \frac{n}{2} - \frac{1}{2}\langle x, y \rangle.$$

Hence, given the inner product $\langle x, y \rangle$ one can recover the Hamming distance, and conversely, via simple affine transformations. This correspondence between the models also modifies the noise parameters, but only in a controlled manner. Since the simulation uses at most two queries and involves only affine transformations, the noise in the simulated model increases by at most a constant multiplicative factor.

$\square$

