# OpenReview forum: "Reconstruction and Secrecy under Approximate Distance Queries"
_NeurIPS.cc/2025/Conference — NeurIPS 2025 spotlight_

### Official Review · Reviewer_M3JV · 2025-07-02

**Clarity:** 4
**Significance:** 3
**Originality:** 3
**Rating:** 4
**Confidence:** 3

**Summary:**

The work studies a reconstruction game between a reconstructor and a responder. The reconstructor can query points in a metric space, and the responder must approximately reveal the distance of the query point to a fixed target point. The goal of the reconstructor is to figure out the location of the target point up to small error. The authors characterize the min-max optimal error in terms of the Chebyshev radius of the metric space. The problem derives some motivation from reconstruction attacks in privacy applications.

**Questions:**

1. Does the main result lead to improved bounds in the context of differential privacy? If not, would it at least give a simpler or cleaner way to prove existing results?
2. In differential privacy applications, the noise scales with the number of queries. In contrast, in the formalism of the paper, the parameters epsilon and delta are fixed. How do you apply the theory to cases where epsilon and delta are a function of the number of queries?
3. Do your results apply to varying epsilon and delta at each step? This could be interesting in applications where different noise levels are used at each step, e.g., gradient-based queries whose norm vary at each update step.

**Ethical Concerns:**

["NO or VERY MINOR ethics concerns only"]

**Final Justification:**

I raised my score based on the authors' discussion with other reviewers.

My enthusiasm is throttled by the lack of learning-theoretic applications, and the lack of any theoretical results about the dependence on the number of steps.

**Limitations:**

Yes.

**Quality:**

3

**Strengths And Weaknesses:**

The paper is well written. The mathematical formulation is elegant and natural. The authors provide ample discussion about the motivation and interpretation of the model.

The main theorem is elegant as it exactly characterizes the error exactly in terms of a natural geometric notion, the Chebyshev radius. Under reasonable additional assumptions this gives upper and lower bounds on the min-max error that match up to constant factor.

There's a nice connection between this problem and Dinur-Nissim style reconstruction problems that are relevant to privacy problems, in particular, differential privacy. However, the paper does not present any new results on privacy. I would've liked to see some new applications either to privacy or other areas where this problem is relevant. The main characterization is nice, but I was hoping to see some new results in specific settings based on this result.

As a consequence, the paper feels a bit more like a SODA/SoCG paper than a NeurIPS paper. Additional connections to machine learning applications would've been nice. The additional results on pseudo-finiteness were a bit too theoretical for my taste as an outsider to this area.

To summarize, this work is an elegant theory paper on a natural question about metric spaces with somewhat limited applications to machine learning.

---

> ### Author Rebuttal · Authors · 2025-07-29
>
> **Response to weaknesses**
>
> We thank the reviewer for the thoughtful and encouraging feedback, and for highlighting the elegance of the model and the main results.
>
> The reviewer raises two main concerns: the lack of concrete applications—especially to privacy—and the paper’s fit to the NeurIPS audience.
>
> We would like to clarify that the primary aim of this work is to identify and study a new interactive learning framework: reconstruction in metric spaces. This abstract formulation captures a variety of contexts, ranging from point-location from noisy signals to Dinur–Nissim–style reconstruction problems relevant to privacy. Within this framework, we analyze fundamental questions about the optimal performance achievable by the reconstructor, establishing baseline results such as Theorem 1, which exactly characterizes the reconstruction error ($\mathrm{OPT}$).
>
> At this early stage, we do not expect these abstract results to yield immediate insights into domain-specific tasks such as privacy-preserving data analysis. That said, we do anticipate such insights to emerge as the theory matures. Moreover, this paper focuses on the reconstructor’s perspective. To study contexts such as private data analysis within this framework, it is essential to consider the responder’s perspective—that is, the agent who answers queries. In privacy-related applications, the responder seeks to provide approximately accurate answers while protecting sensitive data. Just defining what it means for the responder to protect the data, and how to quantify this formally in such a general setting, is itself an interesting challenge. Applications and insights related to differential privacy and private data analysis more broadly are goals we hope to pursue once these foundational questions are addressed. We view the study of the responder’s perspective as an important direction for future work.
>
> *Regarding venue fit:* while our framework naturally encompasses problems of interest to the SoDA and SoCG communities, our approach is grounded in learning theory. The reconstruction game is, at its core, an interactive learning problem, and one of our key contributions is the learning-theoretic characterization of $\mathrm{OPT}$—analogous to Bayes-optimality in standard supervised learning. We therefore believe NeurIPS, with its strong tradition of foundational work in learning theory, is an appropriate venue for this work.
>
> **Responses to questions**
>
> - Question 1: Does the main result lead to improved bounds in differential privacy? If not, does it at least simplify existing proofs?
>
> As noted above, the technical content of this work does not concern differential privacy. Our focus is on analyzing the optimal approximation error from the reconstructor’s perspective. Applications to privacy relate instead to the responder’s perspective—where the goal is to provide approximately correct answers while maintaining uncertainty about the secret point. This aspect is not addressed in the present work, but we view its formalization as an important direction for future research. Once developed, it may lead to meaningful connections with differential privacy.
>
> - Question 2: In differential-privacy applications, noise scales with the number of queries, whereas epsilon and delta are fixed in your formalism. How does the theory apply when epsilon and delta depend on the query count?
>
> This question is closely related to understanding convergence rates to $\mathrm{OPT}(\epsilon, \delta)$. Theorem 1 applies in this setting as well: $\mathrm{OPT}(\epsilon, \delta)$ remains a lower bound even when epsilon and delta depend on the number of queries T. However, the tightness of this bound depends on how fast the reconstructor can approach it. For instance, Theorem 4 shows that in Euclidean spaces of dimension $\ge 2$, this lower bound is never tight unless the noise parameters are zero. As we note in the paper, finding quantitative bounds on these convergence rates in natural metric spaces is an interesting direction for future research. (See also our response to Reviewer 6CpE, where we derive some quantitative bounds that follow from our results.)
>
> - Question 3: Do the results extend to epsilon and delta that vary at each step?
>
> If the sequence of noise levels $(\epsilon_t, \delta_t)$ is non-increasing over time (i.e., the noise weakens or stays constant as the game progresses), then both Theorem 1 as well as Theorem 4 apply with the limiting values of $\epsilon_t$ and $\delta_t$ (with the reservation that if the limit is not attained in finite time, then OPT can also never be attained in a finite number of steps).
>
> For more general sequences, the behavior can be very different. For example, in $\mathbb R^d$, if there are $d + 1$ time steps where both $\epsilon_t$ and $\delta_t$ are zero—and the reconstructor knows which time steps these are—it can locate the secret point exactly by querying an affinely independent set of $d + 1$ points at those times. In such cases, the information available can far exceed what is predicted by our lower bounds.
>
> It would be interesting to study this question further for natural or structured sequences $(\epsilon_t, \delta_t)$, and also in settings where the reconstructor has only partial information or uncertainty about the values of $\epsilon_t$ and $\delta_t$ at each step.

---

> > ### Comment · Area_Chair_Ttdg · 2025-08-06
> > **Discussion required.**
> >
> > Dear Reviewer M3jV,
> >
> > Please read carefully through the authors' responses and check if they address all your concerns.
> >
> > With kind regards,
> >
> > Your AC

---

> > ### Comment · Reviewer_M3JV · 2025-08-07
> >
> > Thanks for your response.
> >
> > My main concern is still with the absence of learning-theoretic applications, which I now believe is somewhat inherent. Since the results are about the case of infinite $T$, the answer is just $(2+\epsilon)\delta$ up to perhaps a factor 2. In most learning-theoretic questions I can think of, however, the *learning* part is about the dependence of the error on the number of steps $T$. In addition, in most applications I can think of (like the Dinur-Nissim style reconstruction bounds) we already have essentially matching upper and lower bounds through various other techniques, like LeCam/Fano for lower bounds, DP mechanisms for upper bounds. (BTW, like the first reviewer, I don't actually see the precise connection here, other than in some moral sense.)
> >
> > To summarize, I think it's a nice geometry fact that the Chebyshev radius can be characterized as this infinite-round point-query game. But I don't see learning-theoretic applications, not even an alternative proof of a known result. There aren't any outlined in the paper, and the discussion initiated by the first reviewer convinced me that even the Dinur-Nissim setting is less related than I first thought.
> >
> > Again, I believe the authors would find a more relevant audience and expert reviewer pool at SocG/SODA. That said, NeurIPS is of course very broad and bar for theory is lower.

---

> > > ### Author Response · Authors · 2025-08-08
> > >
> > > Thank you for your response.
> > >
> > > Regarding the connection to the Dinur–Nissim setting: please see our reply to Reviewer y9Wa, where we explain in more detail how their model arises as a special case of our reconstruction game on the Boolean cube with Hamming distance. We are considering whether to include a more precise explanation of this connection in the paper, and would be glad to hear your thoughts on whether this would be helpful. We would also be grateful if you could refer us to works that obtain tight rates on the approximation error in this setting. We are not aware of any work that studied this setting with multiplicative noise, and the works we know of that consider only the additive case do not provide tight bounds on the approximation factor.
> > >
> > > We’re sorry to hear that you feel the framework lacks learning-theoretic relevance. In our view, the setting is inherently learning-theoretic: it defines a learning framework in which a learner (the reconstructor) aims to approximately learn a hidden point from noisy distance queries. This naturally fits into the well-studied domain of interactive learning. It is also motivated by well-studied applications—ranging from data privacy and group testing to signal recovery—and aims to unify them under a clean learning-theoretic lens.
> > >
> > > Regarding the fact that Theorem 1 improves the guarantee of $(2+\epsilon)\delta$ only by a factor of 2: such refinements are common in learning theory and often mark a meaningful threshold. For example, in classification, 1-nearest neighbor achieves a 2-approximation to the Bayes-optimal error, yet there is a rich literature dedicated to Bayes-consistent algorithms that converge (up to vanishing additive error) to the optimum. In online learning, a factor-2 approximation can be achieved by simple deterministic rules, but much of the field focuses instead on achieving vanishing regret. Similarly, in agnostic PAC learning of halfspaces—a notoriously hard problem—there are efficient algorithms that achieve a constant-factor approximation to the best-in-class error. These are among the most studied topics in learning theory.
> > >
> > > In the same spirit, our work first characterizes the Bayes-optimal reconstruction rate (Theorem 1) and then begins to study how it may be approached. We see this as squarely aligned with central themes in learning theory—perhaps more so than in other areas of TCS, where constant multiplicative factors are sometimes treated as less important. Moreover, even the looser bound of $(2+\epsilon)\delta$ is novel, and we do not see a significantly simpler way to derive it.

---

### Official Review · Reviewer_WJ4i · 2025-07-02

**Clarity:** 4
**Significance:** 3
**Originality:** 4
**Rating:** 5
**Confidence:** 3

**Summary:**

This paper studies what the authors refer to as “the reconstruction game” which is played over a metric space $(X,dist_X)$ between a reconstructor and a responder and is defined as follows: at each time $t$, the reconstructor submits a query $q_t \in X$, and the responder returns a $(1+\epsilon)$-multiplicative, $\delta$-additive “approximate distance” $\hat{d}_t$. After $T$ steps, the feasible region is the set of points in $X$ whose distance to each $q_t$ is approximated by $\hat{d}_t$. At the end of the game, the reconstructor picks a “guess” $x \in X$, and the responder then chooses a final point $y$ in the feasible region (required to be non-empty). The goal of the reconstructor is to carefully choose queries in order to minimize the distance between the final points $x$ and $y$, and the responder is an adversary which is trying to maximize this distance.

The paper is concerned with what is the minimum distance achievable by the reconstructor in the limit as $T \to \infty$. This optimum distance depends on the approximation parameters $\epsilon,\delta$ (the leway afforded to the responder) as well as properties of the metric, name the Chebyshev radius. In particular, their Theorem 1 shows that for many natural metrics, the optimum distance is $\Theta(\delta(2+\epsilon))$.

Next, the authors are interested in understanding the rate of convergence to this optimum value, although this turns out to be challenging. As a first step, they seek to understand for which spaces exact convergence to the optimum occurs after a finite number of steps (finite $T$), and call these spaces “pseudo-finite”. Their second main result (Theorem 4) is that for bounded convex sets in $\mathbb{R}^n$, unless $\epsilon = 0$ or the dimension of the set is $1$, the set is not pseudo-finite.

**Questions:**

Main questions:

What are some of the barriers to understanding the rate of convergence to the optimum? Are there natural special cases or ways to simplify the problem that make it easier?
What happens if the responder is not adversarial? I.e. what if a hidden point is fixed up-front (as in the remark) and the response to each query is some random noisy approximation of the distance? Does this make things significantly easier?


Minor questions, typos, and comments:

There seem to be some numbering discrepancies between the main and full versions. E.g. the full version refers to “Example 6” which I think is supposed to mean “Example 1”.

**Ethical Concerns:**

["NO or VERY MINOR ethics concerns only"]

**Final Justification:**

I believe this paper is well written and contains nice results. I believe it is above the bar for acceptance and I am keeping my score a 5.

**Limitations:**

yes

**Quality:**

3

**Strengths And Weaknesses:**

Strengths: To me, the reconstruction game considered in this work is very natural, and it is surprising to me that it seems like there has been little prior work from a theoretical perspective. Therefore, I consider it to be good in terms of originality. The paper establishes some basic and tight results which seem fundamental. I also think it is well written and clear.

Weaknesses: The paper does not address rate of convergence, which might be more practical and significant, although I recognize this may be very challenging.

---

> ### Author Rebuttal · Authors · 2025-07-30
>
> We thank the reviewer for taking the time to read and appreciate our paper, and for the positive feedback.
>
> We acknowledge the presence of some numbering inconsistencies and will make sure to correct them in the next version of the paper.
> The reviewer also raised a couple of interesting questions that we are happy to address.
>
> **On rates of convergence to $\mathrm{OPT}$ in Euclidean spaces.**
>
> The reviewer was interested in identifying the main obstacles to determining the rate of convergence to $\mathrm{OPT}$. We believe that, in Euclidean spaces, the main challenge lies in proving an upper bound, while the current lower bound in Theorem 4 is likely close to optimal—though a tighter bound may still be possible.
>
> It is important to note that our analysis already yields tight convergence bounds in certain cases. In particular, in the setting of purely multiplicative noise (i.e., $\delta = 0$ and $\epsilon > 0$), our proof shows that convex Euclidean spaces are not pseudo-finite—that is, the approximation error $\mathrm{OPT}_T(\epsilon, \delta)$ does not stabilize after finitely many rounds. A closer inspection of the argument reveals that the convergence rate is at least exponential in this case.
>
> A matching exponential upper bound can be derived using standard space-partitioning techniques from computational geometry: the reconstructor queries a fixed-resolution grid (depending only on the dimension), identifies the grid point with the smallest reported distance, and then refines the search over successively smaller neighborhoods. This strategy yields an exponential decrease in reconstruction error over time.
>
> We will add a remark in the next revision to clarify that our analysis already captures this case.
>
> The case of purely additive noise—i.e., $\delta > 0$ and $\epsilon = 0$—is substantially more subtle. Our proof of Theorem 4 provides a double-exponential lower bound on the convergence rate when the dimension $d > 1$, and it remains open whether this rate is tight. We suspect that it might be, but proving a matching upper bound is challenging. The main difficulty lies in controlling the Chebyshev radius of the feasible region: while the diameter can be reduced relatively easily through adaptive querying, the Chebyshev radius is more sensitive to the internal geometry of the region. This challenge is compounded by the fact that the feasible region is defined as the intersection of differences of balls—a geometrically intricate and non-convex set.
>
> To gain traction on this problem, we are currently investigating a restricted variant in which the reconstructor issues linear queries (which are interesting in their own right).; that is, the reconsturctor selects a direction (i.e. a unit vector) and receives a noisy approximation of the inner product with the secret point. In this setting, the feasible region becomes a convex polytope, making it more amenable to geometric analysis. Interestingly, our characterization of $\mathrm{OPT}$ continues to hold in this linear-query model. We hope that progress in this simplified setting will yield insights applicable to the more general case of distance queries.
>
>
> **On relaxation of the model.**
>
> The reviewer also asked whether the setup becomes easier to analyze—or allows for better approximations—when the responder is not adversarial but instead adds fixed noise to the true answer. This is a very natural and important question.
>
> The answer is yes: when the responder adds a known type of noise in each round—such as independent noise with mean zero—the problem becomes significantly more tractable, and the reconstruction error can decrease substantially. Under suitable assumptions, the error can even converge to zero with high probability.
>
> For instance, in Euclidean space $\mathbb{R}^d$, if each response is perturbed by independent mean-zero noise, the reconstructor can repeat the same query point multiple times and average the responses. This effectively denoises the answers, and by submitting $d+1$ affinely independent queries and averaging appropriately, the reconstructor can approximate the distances—and hence the hidden point—with arbitrarily high accuracy.
>
> This relaxed setting raises several interesting directions for future work, such as studying how different noise distributions affect reconstruction, and how the reconstructor’s level of knowledge (full, partial, or none) about the noise impacts the process. We thank the reviewer for this insightful suggestion and will include it as a proposed direction in the revised version (with an acknowledgment to the reviewer).
>
> Finally, we note that a related variant—discussed earlier in our response on convergence rates—considers linear queries rather than distance queries. In this version, the reconstructor submits directions $v_i$, and the responder returns noisy inner products with the hidden point. This setup leads to a convex feasible region, making it more amenable to analysis.

---

> > ### Comment · Reviewer_WJ4i · 2025-08-01
> > **Rebuttal response**
> >
> > Thank you very much for your detailed and thoughtful responses to my questions! I found them very interesting to read and I think they add a lot of valuable context to the results and problems being considered. I might suggest adding some discussion on these points to the paper, but this is just a suggestion.
> >
> > I have no further questions. Thank you again!

---

> > > ### Author Response · Authors · 2025-08-06
> > >
> > > Thank you for the follow-up! We’re glad you found both the paper and our exchange helpful, and we will include a discussion of the points you raised in the revised manuscript.

---

### Official Review · Reviewer_6CpE · 2025-07-03

**Clarity:** 2
**Significance:** 4
**Originality:** 3
**Rating:** 5
**Confidence:** 2

**Summary:**

This paper studies a two-player reconstruction game to locate an unknown target point $x$ in a metric space via sequential (approximate) distance queries. Concretely, in each round, the reconstructor queries a point $q$ in the space; then the responder replies with a noisy distance $\hat{d}$ between $x$ and $q$, satisfying $(1+\varepsilon)$-multiplicative plus $\delta$-additive distortion bounds. The output of the game is the best reconstructor's worst-case reconstruction error. This paper focuses on studying the rate and limits of such an error as the number of rounds goes to infinity.

**Questions:**

1. In my opinion, the main contribution of this paper is initiating the study on the reconstruction problem from a geometric perspective. Given this, I think it is crucial that the authors clearly discuss all related work and the novelty of the perspective. However, I find that all related works discussed in Appendix A to be not so related to this paper -- in particular, I'm curious whether none of them study the convergence rates of the reconstruction error as in this paper? Moreover, are the authors aware of any additional related work that also studies this in a similar metric-geometry point of view?

2. In the remark on line 77: A Priori vs. A Posteriori Responder, it is stated that the responder waits until *all* queries have been seen before selecting the secret point. Does this mean the constructor has to issue all queries in one batch before seeing any response, i.e. the queries cannot be adaptive? In this sense, the reconstruction game is essentially an one-round game? This confuses me a little since in Section 1.1 Problem Setup, the game is described as being played for T interactive rounds. And reading the following text of the paper, I think the game is still played in T-round. I suggest the authors to clarify the remark a bit to avoid any confusion.

**Ethical Concerns:**

["NO or VERY MINOR ethics concerns only"]

**Final Justification:**

The authors comprehensively answer my questions. The discussion on the convergence rates and multiplicative noise is interesting and insightful. And I agree that "the proof of Theorem 1 is intuitive reflects that it captures the `correct’ characterization."

**Limitations:**

Yes, the limitations are clearly discussed. The authors also point out multiple potential future directions.

**Paper Formatting Concerns:**

A minor thing: I think it is slightly more convenient and conventional if the authors submit the full version of the paper (with the appendix) as the main PDF, as opposed to omitting the appendix in the PDF and only include it in the supplementary material.

**Quality:**

3

**Strengths And Weaknesses:**

**Strength**:

- In my opinion, the authors present this paper as proposing a unified, geometric lens to study the reconstruction problem. I find the viewpoint of interpreting the transcript of the reconstruction game as determining a feasible region interesting; indeed, it elegantly bridges together classical metric‐geometry and the learning problem.

- The main result of this paper (Theorem 1) tightly characterizes the best possible reconstruction error any reconstructor can guarantee in terms of the Chebyshev radius. As the Chebyshev radius is a classical and natural concept in geometry, I find such a characterization nice and clean. In particular, the authors show that the the best attainable reconstruction error is exactly the worst‐case Chebyshev radius over sets of diameter $(2+\varepsilon)\delta$.

**Weakness**:

- Although this paper characterizes the asymptotic optimum of the reconstruction error, it does not provide quantitative rates at which the error converges to its limit, beyond identifying spaces that are "pseudo‐finite" versus those that are not. It would be more satisfying if there's more discussion on the precise finite convergence bounds.

- While this paper discuss the case when the multiplicative error factor $\varepsilon = 0$, it seems that there's no discussion on the case when the additive error $\delta = 0$. In my opinion, the later feels like a more interesting case, as e.g., in approximate similarity search, it is usually more desirable to achieves a multiplicative $(1+\varepsilon)$ error guarantee, as opposed to an additive error guarantee.

- Besides the observation of interpreting the reconstruction game as determining a feasible region, the proof to Theorem 1 feels simple and straight-forward. On the other hand, while the proof ideas to Theorem 4 are more involved, they only lead to results restricted to convex subsets of $\mathbb{R}^n$, leaving the broader landscape of totally bounded metric spaces unsettled.

---

> ### Author Rebuttal · Authors · 2025-07-29
>
> We thank the reviewer for the thoughtful and constructive feedback, and for recognizing the value of our geometric framing and the clean characterization of the reconstruction error in Theorem 1.
>
> **On convergence rates and multiplicative noise**
>
> The reviewer raises two important and closely related points: the lack of explicit discussion on quantitative convergence rates, and the apparent omission of the case of purely multiplicative noise ($\delta = 0$ and $\epsilon >0$). We thank the reviewer for highlighting both issues. While these aspects were not sufficiently emphasized in the current version, our results do apply to the purely multiplicative case and offer insights into convergence rates. We will revise the manuscript to address both points more clearly, as detailed below.
>
> First, we note that Theorem 1 includes the case of $\delta = 0$ as a special case in its characterization of $\mathrm{OPT}$.
> In particular, it implies that $\mathrm{OPT}(\epsilon, 0) = 0$.
>
> As for Theorem 4, while not stated explicitly, our proof yields a tight exponential bound on the convergence rate of $\mathrm{OPT}(\epsilon, 0)$ in the case of purely multiplicative noise (i.e., $\delta = 0$ and $\epsilon >0$). It follows from inspecting our argument that Euclidean spaces are not pseudo-finite: by definition, a space is pseudo-finite if the optimal reconstruction error $\mathrm{OPT}(T, \epsilon, \delta)$ reaches its limiting value in a finite number of rounds. Thus, establishing that Euclidean spaces are not pseudo-finite amounts to proving a non-trivial lower bound on the convergence rate. A closer look at our construction shows that this lower bound is exponential when $\epsilon >0$.
>
> A matching exponential upper bound in the purely multiplicative case follows from standard space-partitioning techniques in computational geometry: the reconstructor can query a uniform grid of fixed size (depending only on the dimension), identify the grid point with the smallest reported distance, and then refine the search by querying a new fixed-size grid centered around that point and scaled to a smaller neighborhood. Repeating this process geometrically shrinks the feasible region, yielding an exponential upper bound on $\mathrm{OPT}(\epsilon, 0)$.
>
> We note in passing that the case of $\epsilon = 0$ and  $\delta > 0$ is more subtle. Our current analysis yields only a double-exponential lower bound on $\mathrm{OPT}$, and it remains open whether this rate is tight.
>
> More broadly, convergence rates outside Euclidean spaces appear to depend heavily on the geometry. We expect that sharp bounds will require case-specific analysis, and we identify this as an important open direction in Question 5.
>
> ⸻
>
> **On the proof of Theorem 1**
>
> While we agree that the proof of Theorem 1 is intuitive once the right definitions are in place, we respectfully note that this intuitiveness reflects the fact that Theorem 1 captures the `correct’ characterization. As is often the case in mathematics, a proof may appear straightforward once the key concepts are established—but discovering those concepts can require significant insight.
>
> In our case, it took considerable effort to identify what governs $\mathrm{OPT}$ in this reconstruction game. Two key observations guided us: (i) if the reconstructor is allowed to query all points in the space, then the feasible set has diameter at most $\delta\cdot(\epsilon+2)$, and (ii) the responder can maintain any set of that diameter as feasible for all $T$. These two facts pointed toward the Chebyshev radius as the central quantity, and once this connection was made, the form of the characterization followed naturally. Our effort in uncovering this structure is further reflected in the wide range of metric spaces we analyzed—both to test the sharpness of our characterization and to understand its implications. We included over five distinct examples in the paper, spanning diverse geometries, and discussed others along the way.
>
> The proof may now seem smooth, but this reflects the fact that Theorem 1 captures the correct structure of the problem. We hope the reviewer will recognize the mathematical effort and non-triviality involved in identifying the right characterization.
>
> ⸻
>
> **On related work**
>
> The reviewer raises an important point regarding the absence of references on convergence rates for the reconstruction error $\mathrm{OPT}(\epsilon, \delta)$. We emphasize that this reflects a genuine gap in the literature rather than an oversight. While preparing the paper, we searched extensively for prior work addressing convergence rates in the general setting we consider and were surprised to find none.
>
> In particular, we consulted several experts across both broad areas—such as theoretical computer science, combinatorics, and geometry—and more specific fields, including learning theory, computational geometry, and differential privacy. These conversations confirmed not only the lack of direct prior work on the problem, but also that the question appears natural and its absence in the literature is somewhat surprising. We also conducted extensive literature searches using academic databases and large language models (e.g., ChatGPT and Gemini). Throughout the development of the paper, we continuously refined our Google searches with updated keywords—for example, after identifying the relevance of the Chebyshev radius and Jung’s constant, we included these terms in our queries.
>
> These efforts did not uncover prior work directly addressing convergence rates of $\mathrm{OPT}(\epsilon, \delta)$, though they did bring to our attention related but more specialized notions. For example, we discovered the notion of sequential dimension in graphs through these searches. In contrast, our work was directly inspired by earlier reconstruction attacks in differential privacy, such as the Dinur–Nissim framework, which we were already aware of. We discuss both of these connections, as well as others, in Appendix A.
>
> ⸻
>
> **On the responder model and adaptivity**
>
> The reviewer asks for clarification regarding the remark on line 77 about the “a posteriori” responder, which states that the responder selects the secret point only after all queries have been issued. The concern is whether this implies that the reconstructor must submit all queries at once (i.e., non-adaptively), which seems inconsistent with the T-round interactive game described in Section 1.1.
>
> We thank the reviewer for pointing out this potential confusion. To clarify: in our formulation, both the reconstructor and the responder are allowed to act adaptively. The game proceeds for T rounds, during which the reconstructor chooses each query based on past responses, and the responder chooses each response based on all queries and its chosen secret point.
>
> The distinction between a priori and a posteriori responder models concerns when the responder commits to the secret point. In the a priori model, the responder selects the secret point before seeing the queries. In the a posteriori model, the responder waits until all queries are revealed before choosing the secret point, giving it an additional degree of adaptivity. This latter setting is the one we focus on in the paper.
>
> We will revise the remark to clarify this point and avoid confusion.

---

> > ### Comment · Reviewer_6CpE · 2025-08-04
> >
> > Thank you for your detailed response and for the comprehensive answers to my questions! The discussion on the convergence rates and multiplicative noise is interesting and insightful. And I completely agree that "the proof of Theorem 1 is intuitive reflects that it captures the `correct’ characterization.'' I have no further questions and have adjusted my rating to 5.

---

> > > ### Author Response · Authors · 2025-08-05
> > >
> > > Thank you for the follow-up; we’re pleased that our clarifications addressed your concerns. We appreciate you taking the time to revisit the manuscript and raise the rating to 5, and we will use your feedback to further refine the paper.

---

### Official Review · Reviewer_y9Wa · 2025-07-23

**Clarity:** 4
**Significance:** 2
**Originality:** 3
**Rating:** 5
**Confidence:** 3

**Summary:**

This paper considers the problem of guessing an unknown point in a metric space,  using a series of noisy distance queries. More specifically, the paper considers a "reconstruction game", which involves two players: a reconstructor, who aims to accurately guess the target point, and a responder, who provides approximate distances under additive and multiplicative noise. The main goal is to study the limit of the reconstruction error in the worst case (i.e., for the most difficult point to reconstruct), assuming that the responder knows all the queries in advance, or that the recontstructor is "deterministic" (which, I suppose, means "non interactive", i.e., cannot change the queries depending on previous answers).

The paper offers two main contributions:

- A characterization of the optimal approximation error that the reconstructor can obtain as the number of queries tends to infinity. This limit is shown to be related to the Chebyshev radius.

- A characterization of the pseudo-finite metric spaces, which enjoy the property that the optimal reconstruction error can be achieved after a finite number of queries.  The paper shows that for bounded convex subsets of the Euclidean space, a space is pseudo-finite if and only if its dimension is one and there is no multiplicative noise.

**Questions:**

When you say "deterministic reconstructor", you mean "non interactive"?

**Ethical Concerns:**

["NO or VERY MINOR ethics concerns only"]

**Final Justification:**

After the exchange with the authors, I can see now how their approach can apply to privacy attacks like the one of  Dinur and Nissim. I understand now the power and practical applicability of the distance-based games taht the authors are proposing, and for this reason, I have raised my score.

**Limitations:**

I would recommend the authors to provide at least one detailed practical example where the problem they propose is relevant.

**Paper Formatting Concerns:**

I did not find any issue

**Quality:**

3

**Strengths And Weaknesses:**

## Strengths

- The paper addresses a cute and intriguing theoretical problem, and the results are interesting

- The paper is well written and seems technically solid.

## Weaknesses

- Although I like the paper, I am concerned about its practical relevance.  I am not convinced by the discussion provided in the introduction to justify the importance of the problem addressed in various fields. The relation with differential privacy (which is a topic I am very familiar with) seems to me rather contrived: I cannot imagine a scenario in which all the queries are fixed in advance, and, more importantly, the noise of the responder would need to adapt due to the consumption of the privacy budget. The paper also lists a series of other areas (for instance, computational geometry), for which the paper claims that the problem is relevant. On these, I cannot say for sure, because I am not an expert, but the explanation is very terse.   I would have appreciated it better to see at least one detailed practical example where the problem is relevant.

---

> ### Author Rebuttal · Authors · 2025-07-29
>
> We thank the reviewer for the thoughtful feedback and for finding the problem intriguing and the paper well-written. We now address the concerns regarding adaptivity and practical relevance.
>
> Regarding the reviewer’s question: our use of the term “deterministic reconstructor” may have caused confusion. To clarify: deterministic does not mean non-adaptive (or oblivious). In our setting, the reconstructor is fully adaptive—it chooses each query based on the interaction up to that point. The qualifier “deterministic” simply means that the query at each time step is a deterministic function of the history so far, as opposed to being sampled at random. We focus on this case primarily for simplicity of presentation; however, our results and arguments extend naturally to the more general case of randomized reconstructors as well. We will revise the paper to clarify this point.
>
> As the reviewer noted, this misunderstanding may have contributed to the impression that the paper has limited relevance to practical problems. We hope that, after clarifying this point, the broader scope of our framework becomes more apparent.
> In addition, while our model is indeed theoretical, it is grounded in and generalizes concrete setups with clear real-world impact. One prominent example is the reconstruction attacks of Dinur and Nissim, which played a central role in the definition of differential privacy. These attacks correspond to a special case of our model on the Boolean cube with Hamming distance. Our framework captures such attacks and extends them to a broader class of metric spaces and query types. Another example is point-location problems arising in signal processing and sensor networks, where one aims to localize a signal source based on noisy measurements. Such problems also fit naturally into our model and were a motivating example in developing it.
>
> We note, however, that at this stage, we do not expect nor aim for our abstract results to yield immediate insights into domain-specific settings. Rather, our hope is that the framework we propose will serve as a convenient, clean, and powerful tool for understanding and reasoning about problems arising in important application domains. For example, it would be interesting to study private data analysis in this setting—namely, to what extent the responder can retain uncertainty about its secret input point $x$ while still providing sufficiently accurate answers. Quantifying such uncertainty in a geometric context is a compelling and nontrivial challenge, and we hope that future work will develop this direction further. In contrast, the theorems in the present work are centered on the reconstructor’s perspective and aim to establish basic capabilities and limitations of learning from noisy metric feedback.
>
> To summarize, this is indeed a theoretical contribution, but one that unifies—through a learning-theoretic formulation—a variety of reconstruction problems that have both been studied in practice and shown to be of significant practical impact. We believe and hope that by distilling the common structure underlying these problems, our framework provides a useful foundation for future theoretical and applied work alike.

---

> ### Comment · Reviewer_y9Wa · 2025-08-01
>
> I would like to thank the authors for their reply. I understand better now the connection between the problem they consider in their paper and privacy. I was confused because the inequalities at Line 22 in the introduction resemble those of differential privacy, and I thought--mistakenly--that their motivation in terms of privacy referred to this analogy.
>
> I also appreciate the further justification they give in the rebuttal about the potential impact of their contribution, and I invite them to add that justification in the introduction of their paper.

---

> ### Comment · Reviewer_y9Wa · 2025-08-01
>
> A minor point:  I don't completely see how Dinur and Nissim's reconstruction attack is a special case of their problem on the Boolean cube with Hamming distance:
>
> I agree that Dinur and Nissim model a private database as a point sequence on the Boolean cube, say $D=(d_1,\dots,d_n)$, where each $d_i$ is a bit. However, as far as I remember, a query is determined by a subset of indexes $S = $ {${i_1},\dots, {i_k} $}  representing a subspace and not a point. As for the distance, define $q_S(D)=\sum_{i\in S}d_i$. Given an approximate answer $a$ to the query, the distance they consider is  $|a-q_{S}(D)|$, which is less discriminative than the Hamming distance.
>
> Anyway, I agree that the analogy is there, and perhaps a similar reconstruction attack can be done using points and the Hamming distance.

---

> > ### Author Response · Authors · 2025-08-05
> >
> > We appreciate the reviewer’s thoughtful feedback and the acknowledgement that our clarification strengthens the paper’s privacy relevance. In light of this added significance, we would be grateful if the reviewer could kindly reconsider the numerical score. Of course, we fully respect the reviewer’s discretion and thank them again for their careful evaluation.
> >
> > We also wanted to clarify that Dinur–Nissim-style reconstruction attacks are in fact essentially equivalent to our distance-based game. We briefly discuss this connection in Appendix~A (Related Work), where we sketch how counting queries can be translated into distance queries and vice versa. If that explanation wasn’t sufficiently clear, we’d be happy to expand on it.

---

> > > ### Comment · Reviewer_y9Wa · 2025-08-06
> > >
> > > Dear authors, can you please provide a hint of the proof of equivalence that you mentioned in your last comment? (citing from the comment: "Dinur–Nissim-style reconstruction attacks are in fact essentially equivalent to our distance-based game"). Just a hint, you don't need to go into detail.
> > >
> > > And to answer your request: yes, I will consider increasing the numerical score.

---

> ### Comment · Reviewer_y9Wa · 2025-08-06
>
> I have re-read Appendix A, but I still do not understand the explanation of the equivalence. Specifically, it seems to me that the queries considered by Dinur and Nissim are different than the distance queries, because they are counting queries on a subset of indexes, whereas (if I understand correctly) your transformation is between points in the full n-dimensional space.
> Are you saying, perhaps, that the correspondence is not one-to-one? I.e., a Dinur and Nissim counting query corresponds to a set of distance queries (not just one distance query)?

---

> ### Author Response · Authors · 2025-08-07
>
> Thank you for your interest in these details. It made us consider adding a more formal and detailed discussion of how the Dinur–Nissim counting-query game can be seen as a special case of our reconstruction game. We’d be glad to hear your thoughts on whether you think this would be a useful addition.
>
> The counting-query game in Dinur–Nissim’s model is equivalent to the distance-based game played on the Boolean cube with the Hamming metric, in the sense that every query in one game can be simulated by at most two queries in the other.
> In a nutshell, this follows because the squared $\ell_2$-distance between two Boolean vectors in $\{\pm 1\}^n$ is equal to four times their Hamming distance, and the latter is an affine function of their inner product:
>
> $$\mathrm{dist}_{\text{Ham}}(x, y) = \frac{1}{4} \|x - y\|_2^2 = \frac{n}{2} - \frac{1}{2} \langle x, y \rangle.$$
>
> To show in more detail that the counting-query game is equivalent to the distance-based game on the Boolean cube (with Hamming distance), we introduce an intermediate step: both games are equivalent to an inner-product game played on $\{\pm 1\}^n$. The intermediate inner-product game is defined as follows: the responder selects a secret vector $D’ = (d_1’, \ldots, d_n’) \in ֿ{\pm 1\}^n$. In each round, the reconstructor submits a query vector $w = (w_1, \ldots, w_n) \in \{\pm 1\}^n$, and the responder returns a noisy approximation of the inner product:
>
> $$\langle D’, w \rangle = \sum_{i=1}^n w_i \cdot d_i’.$$
>
> **Step 1: From the Dinur–Nissim model to the inner-product game**
>
> In the Dinur–Nissim model, the dataset is a binary vector $D = (d_1, \ldots, d_n) \in {0,1}^n$, and each query is a subset $q \subseteq [n]$, whose answer is the count
>
> $$a_q = \sum_{i \in q} d_i.$$
>
> We can represent the subset $q$ by its indicator vector $v_q \in \{0,1\}^n$, so that $a_q = \langle D, v_q \rangle$.
>
> To simulate this count using the inner-product game on $\{\pm 1\}^n$, consider the transformation
>
> $$v \mapsto 2v - 1,$$
>
> which maps $\{0,1\}^n$ to $\{\pm 1\}^n$. Let
>
> $$D’ = 2D - 1 \quad \text{and} \quad w_q = 2v_q - 1.$$
>
> Then we have the identity
>
> $$\langle D’, w_q \rangle = 4 \langle D, v_q \rangle - 2 \langle D, \mathbf{1} \rangle - 2|q| + n.$$
>
> Therefore, we can recover the original count $\langle D, v_q \rangle$ by submitting two inner-product queries: one with $w_q$ and one with the all-ones vector $\mathbf{1}$. A similar argument gives the reverse direction.
>
> **Step 2: From the inner-product game to the distance-based game**
>
> Next, we show that the inner-product game on $\{\pm 1\}^n$ is equivalent to the distance-based game on $\{\pm 1\}^n$ with the Hamming metric. As noted earlier:
>
> $$\mathrm{dist}_{\text{Ham}}(x, y) = \frac{1}{4} \|x - y\|_2^2 = \frac{n}{2} - \frac{1}{2} \langle x, y \rangle.$$
>
> So given the inner product $\langle x, y \rangle$, we can compute the Hamming distance, and vice versa, using only affine transformations.
>
> We remark that this transformation between the models also affects the noise parameters in a controlled way. Since the simulations involve only simple affine transformations and at most two queries, the resulting noise in the simulated model changes by at most a constant multiplicative factor.

---

> > ### Comment · Reviewer_y9Wa · 2025-08-08
> >
> > Thank you very much for the detailed explanation! It's very interesting that you can transform the Dinur–Nissim query game into your reconstruction game based on distance. I suggest including the details of the transformation intro your revised paper, at least in the supplementary material, because it's very inspiring.
> >
> > I will increase my score. Best regards.

---

### Note · Authors · 2025-08-16

Dear Area Chairs and Reviewers,

Thank you for taking the time to read our paper and for providing thoughtful and constructive feedback throughout the review process.

We would like to summarize the rebuttal process for clarity. The initial reviewer scores were 3, 4, 4, and 5. All reviewers found the mathematical learning model elegant, and some raised questions regarding related work, quantitative bounds, other technical details, and applicability in differential privacy. During the rebuttal, we did our best to address these points, and the reviewers who initially gave scores of 4 indicated they would raise their evaluations. In our final exchange with the remaining reviewer, we hope our clarifications were helpful, and we would be grateful if they might also consider raising their score.

Following the fruitful exchange with the reviewers, we plan to incorporate several clarifications into the final version. For example:

1.	Clarifying the connection between our reconstruction-game framework and the Dinur–Nissim model — explicitly stating how the Dinur–Nissim reconstruction attack corresponds to a special case of our framework on the Boolean cube with Hamming distance, thus establishing the equivalence.

2.	Explaining more precisely what we mean by a “deterministic” reconstructor in our setting — namely, adaptive but non-randomized — and noting that in both responder models, the game remains fully interactive.

We again thank the reviewers and Area Chairs for their constructive engagement with our work.

Best regards,

The Authors

---

### Decision · Program_Chairs · 2025-09-17

**Decision:**

Accept (spotlight)

**Comment:**

This paper considers a "reconstruction game", which involves two players: a reconstructor, who aims to accurately guess the target point, and a responder, who provides approximate distances under additive and multiplicative noise. The main goal is to study the limit of the reconstruction error in the worst case (i.e., for the most difficult point to reconstruct), assuming that the responder knows all the queries in advance, or that the recontstructor is "deterministic".

Strengths: This paper characterizes the optimal approximation error that the reconstructor can obtain as the number of queries tends to infinity. This limit is shown to be related to the Chebyshev radius. The paper  shows that in the pseudo-finite metric spaces the optimal reconstruction error can be achieved after a finite number of queries.

There are some concerns about the practicality of the paper, the reviewers agree that this is an exciting result for a novel formulation of a problem of interest.